# Enterobacterales plasmid sharing amongst human bloodstream infections, livestock, wastewater, and waterway niches in Oxfordshire, UK

William Matlock[1]*[†], Samuel Lipworth[1,2][†], Kevin K Chau[1], Manal AbuOun[3], Leanne Barker[1], James Kavanagh[1], Monique Andersson[2], Sarah Oakley[2], Marcus Morgan[2], Derrick W Crook[1,2,4], Daniel S Read[5], Muna Anjum[3], Liam P Shaw[6][‡], Nicole Stoesser[1,2,4]*[‡], REHAB Consortium

[1]Nuffield Department of Medicine, University of Oxford, Oxford, United Kingdom; [2]Oxford University Hospitals NHS Trust, Oxford, United Kingdom; [3]Animal and Plant Health Agency, Addlestone, United Kingdom; [4]NIHR Biomedical Research Centre, Oxford, United Kingdom; [5]Centre for Ecology and Hydrology, Wallingford, United Kingdom; [6]Department of Biology, University of Oxford, Oxford, United Kingdom

**\*For correspondence:**
william.matlock@ndm.ox.ac.uk
(WM);
nicole.stoesser@ndm.ox.ac.uk
(NS)

[†]These authors contributed equally to this work
[‡]These authors also contributed equally to this work

**Group author details:**
REHAB Consortium See page 17

**Competing interest:** The authors declare that no competing interests exist.

**Abstract** Plasmids enable the dissemination of antimicrobial resistance (AMR) in common Enterobacterales pathogens, representing a major public health challenge. However, the extent of plasmid sharing and evolution between Enterobacterales causing human infections and other niches remains unclear, including the emergence of resistance plasmids. Dense, unselected sampling is essential to developing our understanding of plasmid epidemiology and designing appropriate interventions to limit the emergence and dissemination of plasmid-associated AMR. We established a geographically and temporally restricted collection of human bloodstream infection (BSI)-associated, livestock-associated (cattle, pig, poultry, and sheep faeces, farm soils) and wastewater treatment work (WwTW)-associated (influent, effluent, waterways upstream/downstream of effluent outlets) Enterobacterales. Isolates were collected between 2008 and 2020 from sites <60 km apart in Oxfordshire, UK. Pangenome analysis of plasmid clusters revealed shared 'backbones', with phylogenies suggesting an intertwined ecology where well-conserved plasmid backbones carry diverse accessory functions, including AMR genes. Many plasmid 'backbones' were seen across species and niches, raising the possibility that plasmid movement between these followed by rapid accessory gene change could be relatively common. Overall, the signature of identical plasmid sharing is likely to be a highly transient one, implying that plasmid movement might be occurring at greater rates than previously estimated, raising a challenge for future genomic One Health studies.

## Editor's evaluation

This study presents valuable findings on the dissemination of plasmids. The analysis, involving a geographically and temporally restricted collection of fully assembled genomes of 1458 isolates carrying in total of 3697 plasmids representing five major Enterobacterales genera, convincingly demonstrated that similar plasmids were shared between genera, species, and clones, within and between ecological niches. Given the size of the dataset and the very detailed level of analysis, this important study contributes to insights into the flow of plasmids, including those carrying antimicrobial resistance genes, across niches.

## Introduction

Enterobacterales are found both in human niches (e.g. hospital patients [*Linh et al., 2021*; *Kraftova et al., 2021*] and wastewater [*Cahill et al., 2019*]) and in non-human niches (e.g. livestock-associated [*Subramanya et al., 2021*; *AbuOun et al., 2021*] and waterways [*Díaz-Gavidia et al., 2021*]). In recent decades, widespread carriage of antimicrobial resistance (AMR) genes has complicated the treatment of Enterobacterales infections (*Buchy et al., 2020*; *Ruppé et al., 2020*). The dissemination of AMR genes between Enterobacterales occurs in a 'Russian-doll'-style hierarchy of nested, mobilisable genetic structures (*Sheppard et al., 2016*): genes not only move between bacterial hosts on mobilisable or conjugative plasmids but can also be transferred within and between plasmids and chromosomes by smaller mobile genetic elements (MGEs) such as insertion sequences (*Che et al., 2021*; *Shaw et al., 2021*). Despite gene gain/loss events, many plasmids have been shown to have a persistent structure encoding replication and transfer machinery (*Orlek et al., 2017b*; *Matlock et al., 2021a*).

Many plasmids can transfer between species and are seen across different niches (*Redondo-Salvo et al., 2020*) but the extent to which they are shared between human and non-human niches remains poorly understood. Previous studies investigating this topic have often been limited in size given the genetic diversity in these niches (*Mounsey et al., 2021*), and/or restricted to single species (*Ludden et al., 2019*) or drug-resistant isolates (*Shen et al., 2020*), or are systematic studies, pooling geographically/temporally disparate samples (*Cherak et al., 2021*; *Bastidas-Caldes et al., 2022*). Further, fragmented genome assemblies in many cases make recovering complete plasmids, and other MGEs, impossible (*Hilpert et al., 2021*).

Instances of cross-niche transfer of plasmids are well described, but the frequency of such events is poorly characterised. There are multiple instances where AMR genes have emerged from non-human niches and subsequently become major clinical problems in human Enterobacterales infections, highlighting the relevance of inter-niche transfer in AMR gene dissemination (e.g. $bla_{CTX-M}$, $mcr$-1 [*Wang et al., 2018*] and $bla_{NDM-1}$ [*Sekizuka et al., 2011*]). In general, environmental bacteria are believed to be the original source of AMR genes that eventually become prevalent in clinical settings after transfer into clinical pathogens. However, we know little about natural rates of inter-niche transfer beyond these high-profile examples. It remains unclear how plasmids evolve within natural populations, meaning we understand little about the wider context in which AMR genes emerge and disseminate.

To explore Enterobacterales plasmid diversity and sharing across niches in a geographically and temporally restricted context, we studied hybrid assemblies (i.e. using both long and short reads) of large Enterobacterales isolate collections in Oxfordshire, UK, from (i) human bloodstream infections (BSIs; 2008–2018), (ii) livestock-associated sources (faeces from cattle, pigs, poultry, sheep; surrounding environmental soils; all 2017 except poultry 2019–2020), and (iii) wastewater treatment work (WwTW)-associated sources (influent, effluent, waterways upstream/downstream of effluent outlets; Oxfordshire, 2017).

## Results

Our dataset of *n=3697* plasmids from *n=1458* isolates (*Figure 1a*, *Table 1*) contained bacteria from human BSIs (*n=1880* plasmids from *n=738* isolates), livestock-associated sources (cattle, pig, poultry, and sheep faeces, soils surrounding livestock farms; *n=1155* plasmids from *n=512* isolates), and from wastewater treatment works (WwTW)-associated sources (influent, effluent, waterways upstream/downstream of effluent outlets; *n=662* plasmids from *n=208* isolates). All sampling sites were <60 km apart (*Figure 1b*) and timeframes overlapped (2008–2020; *Figure 1c*). Isolates had a median 2 plasmids (IQR = 1–4, range = 0–16). Major Enterobacterales genera represented included: *n=1044 Escherichia*, *n=212 Klebsiella*, *n=125 Citrobacter*, and *n=63 Enterobacter*.

Sampling niche was strongly associated with isolate genus (Fisher's test, p-value <0.001; *Table 2*). *Klebsiella* isolates were disproportionately derived from BSI versus other niches (22% [161/738] *Klebsiella* from BSI versus 8% [51/669] from other niches). *Citrobacter* and *Enterobacter* were disproportionately derived from WwTW-associated versus other niches (51% [107/208] *Citrobacter* and *Enterobacter* from WwTW versus 6% [81/1250] *Citrobacter* and *Enterobacter* from other niches). Chromosomal Mash trees (see Materials and methods) for the two most common species in the dataset, *Escherichia coli* (72% [1,044/1,458]; see *Appendix 1—figure 1*) and *Klebsiella pneumoniae* (11% [163/1458];

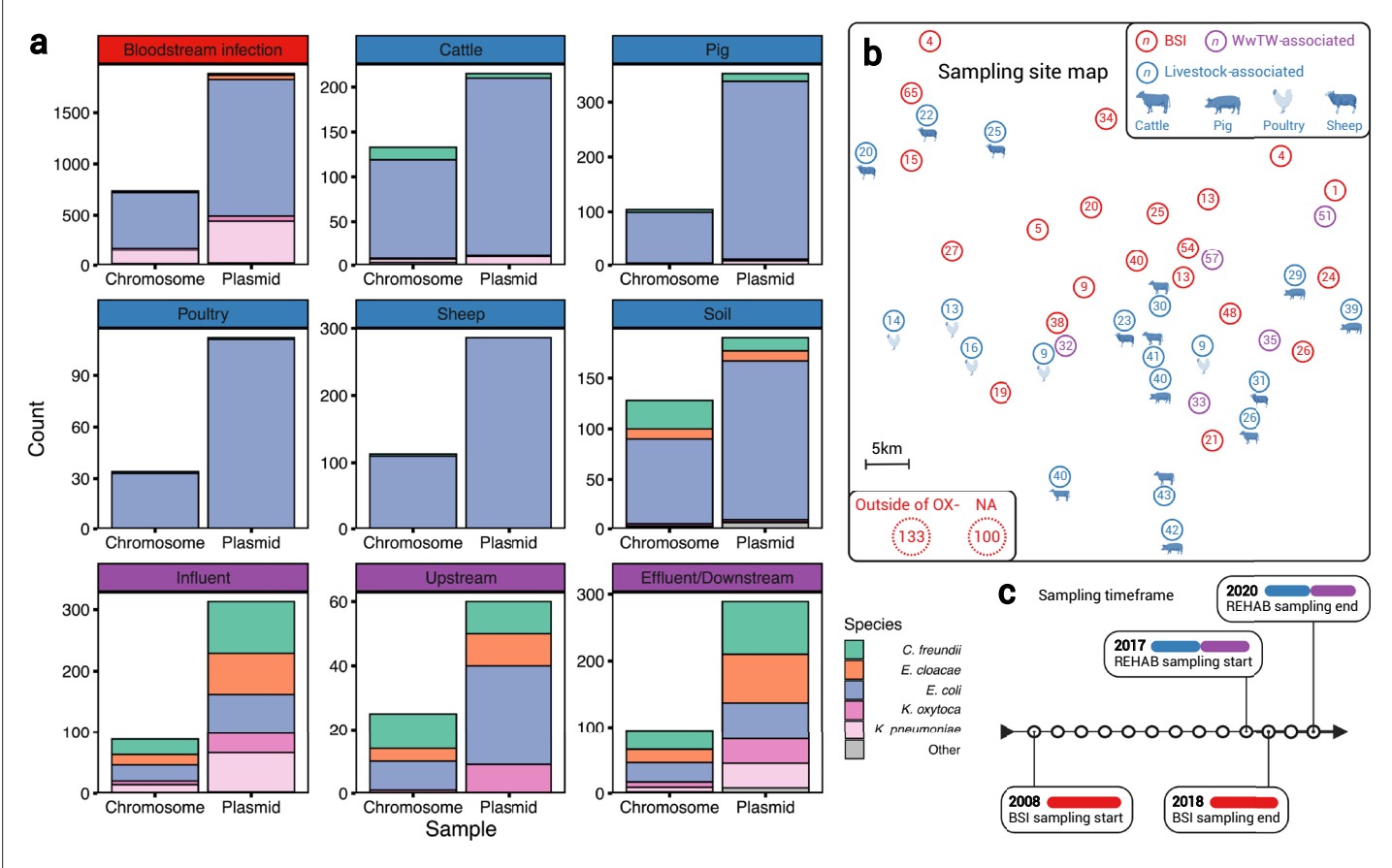

**Figure 1.** A diverse sample of geographically and temporally restricted Enterobacterales. (**a**) Number of chromosomes and plasmids by niche, stratified by isolate genus. (**b**) Map of approximate, relative distances between sampling sites, coloured by niche (human bloodstream infection [BSI], livestock-associated [cattle, pig, poultry, and sheep faeces, soils nearby livestock sites], and wastewater treatment work [WwTW]-associated sources [influent, effluent, waterways upstream/downstream of effluent outlets]). Number in circles indicates how many of the *n=1458* isolates are from that location. (**c**) Sampling timeframe for BSI and REHAB (non-BSI) isolates.

**Table 1.** Isolate niche breakdown.

| Niche | Sample type(s) | No. isolates | No. plasmids |
|---|---|---|---|
| Bloodstream infections (BSIs) | Community, nosocomial, and other healthcare-associated infections | 738 | 1880 |
| | Cattle faeces | 133 | 215 |
| | Sheep faeces | 113 | 286 |
| | Pig faecesan | 104 | 352 |
| | Poultry faeces | 34 | 112 |
| Livestock-associated | Soil surrounding livestock farms | 128 | 190 |
| | Influent | 88 | 313 |
| | Upstream waterways | 25 | 60 |
| Wastewater treatment work (WwTW)-associated | Effluent and downstream waterways | 95 | 289 |
| Total | | 1458 | 3697 |

**Table 2.** Isolate genus breakdown.

| Niche | Isolate genus | | | | | |
|---|---|---|---|---|---|---|
| | *Citrobacter* | *Enterobacter* | *Escherichia* | *Klebsiella* | *Other* | **Total** |
| Bloodstream infections (BSIs) | 6 | 11 | 547 | 161 | 13 | **738** |
| Livestock-associated | 54 | 10 | 433 | 14 | 1 | **512** |
| Wastewater treatment work (WwTW)-associated | 65 | 42 | 64 | 37 | 0 | **208** |
| Total | 125 | 63 | 1044 | 212 | 14 | 1458 |

*Appendix 1—figure 2*), demonstrated intermixing of human and non-human isolates within clades, consistent with species lineages not being structured by niche.

We contextualised our plasmids within known plasmid diversity using 'plasmid taxonomic units' (PTUs; using COPLA, see Materials and methods), designed to be equivalent to a plasmid 'species'. We found 32% (1193/3697) of plasmids were unclassified, highlighting the substantial plasmid diversity within this geographically restricted dataset, whilst the remaining 68% (2,504/3,697) were assigned a PTU. In total, we found $n=67$ known PTUs, containing a median 9 plasmids (IQR = 4–30, range = 1–556), with the largest PTU-$F_E$ (556/2,504), corresponding to F-type *Escherichia* plasmids.

## Near-identical plasmid sharing observed between human and livestock-associated Enterobacterales

We screened for near-identical plasmids shared across isolates by grouping those with a low Mash distance ($d<0.0001$) and highly similar lengths (longest plasmid ≤1% longer than shorter plasmids; note that this near-identical threshold becomes an identical threshold for extremely small plasmids; see Materials and methods). We found $n=225$ near-identical groups of ≥2 members, recruiting 19% (712/3697) plasmids. Bootstrapping accumulation curves for near-identical plasmid groups and singletons per the number of isolates (ACs; see Materials and methods), we revealed a highly 'open' accumulation (Heap's parameter $\gamma=0.97$, *Appendix 1—figure 3*), suggesting further isolate sampling would detect more unique plasmids approximately linearly. Restricted to BSI/livestock-associated isolates alone, we found similar curves for both niches (BSI $\gamma=0.98$, livestock-associated $\gamma=0.94$), suggesting they had similar levels of plasmid diversity.

The most common group size of near-identical plasmids were pairs, representing 71% (159/225) of groups (group size IQR = 2–3, range = 2–32). Plasmid members of near-identical groups represented multiple bacterial host STs (25% [56/225]), species (4% [9/225]), and genera (4% [9/225]), consistent with plasmids capable of inter-lineage/species/genus transfer. Further, 8% (17/225) of near-identical groups contained plasmids found across human BSIs and at least one other sampling niche (livestock-associated/WwTW-associated), suggesting inter-niche transfer (i.e. 'cross-niche groups'; *Figure 2a*). Within cross-niche groups, $n=3/17$ contained plasmids from multiple bacterial species (*Figure 2b*), and most consisted of conjugative plasmids ($n=5/17$ conjugative, $n=9/17$ mobilisable, $n=3/17$ non-mobilisable; *Figure 2c*). AMR genes were carried by plasmids in $n=6/17$ cross-niche groups (*Figure 2d*), with $n=5/6$ of these groups containing at least one beta-lactamase protein encoding gene.

Sharing between BSI and livestock-associated isolates was supported by 8/17 cross-niche groups ($n=45$ plasmids). Of these, $n=3/8$ groups contained BSI/sheep plasmids: one group contained mobilisable Col-type plasmids, the remaining two groups contained conjugative FIB-type plasmids, of which one group contained plasmids carrying the AMR genes *aph(3'')-Ib*, *aph(6)-Id*, *bla*$_{TEM-1}$, *dfrA5*, *sul2*, and the other group contained plasmids carrying the MDR efflux pump protein *robA* (see Materials and methods). A further $n=2/8$ groups contained BSI/pig mobilisable Col-type plasmids, of which one group other carried the AMR genes *aph(3'')-Ib*, *aph(6)-Id*, *dfrA14*, and *sul2*. Lastly, $n=1/8$ groups contained BSI/poultry non-mobilisable Col-type plasmids, $n=1/8$ contained BSI/pig/poultry/influent non-mobilisable Col-type plasmids, and $n=1/8$ contained BSI/cattle/pig/poultry/influent mobilisable Col-type plasmids.

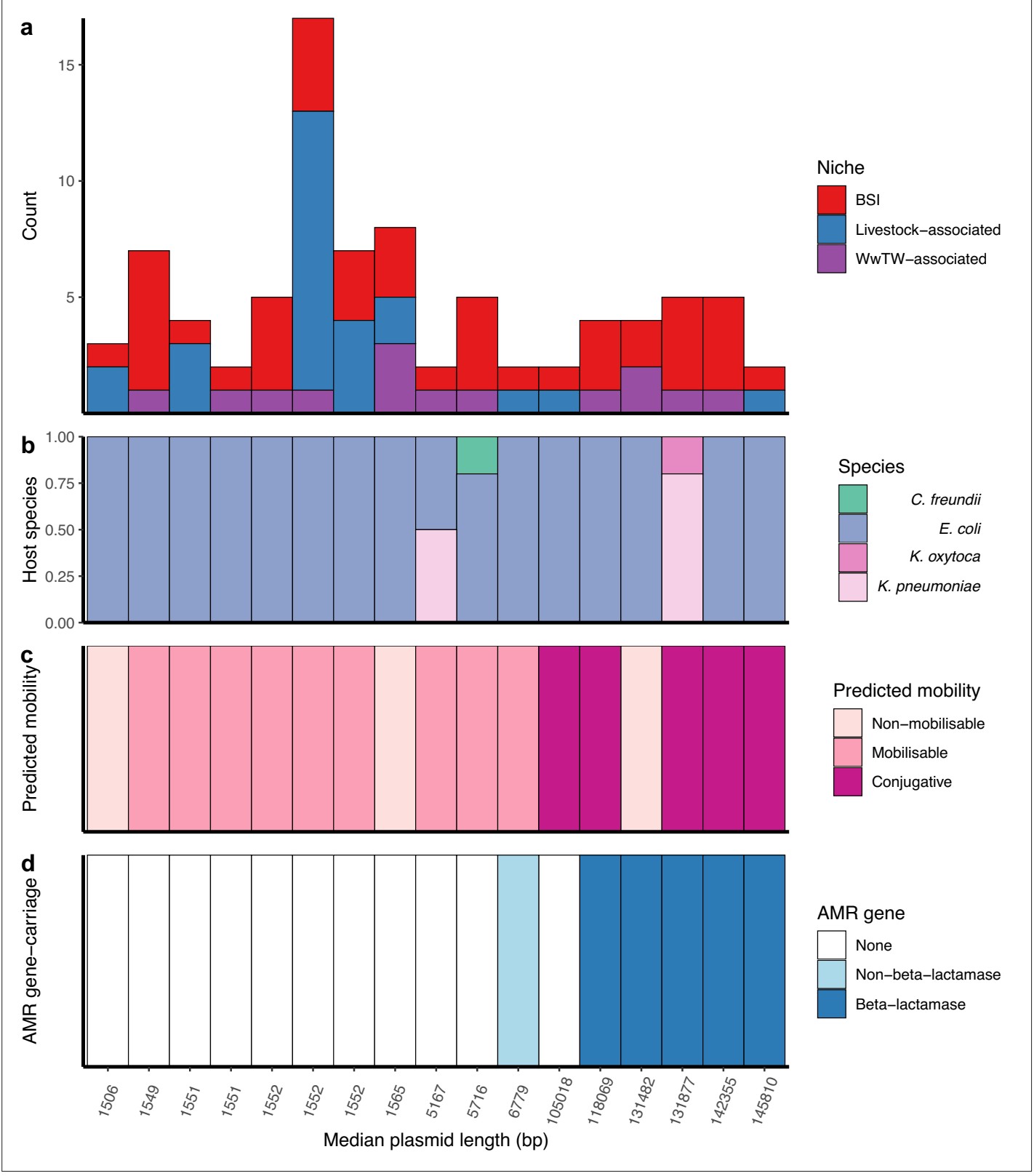

**Figure 2.** Cross-niche, near-identical plasmids. (**a**) Size of cross-niche, near-identical plasmid groups, coloured by niche (total *n=84* plasmids). Median length (bp) of plasmids within groups increases from left to right. (**b**) Proportion of plasmid host species by group. (**c**) Predicted mobility of plasmid. (**d**) Antimicrobial resistance (AMR) gene carriage in plasmid. For small plasmids, the stringent distance threshold (*d*<0.0001) becomes an identical threshold, meaning that plasmids of the same length with a single SNP between them are grouped into different groups (e.g. the three groups with

*Figure 2 continued on next page*

*Figure 2 continued*

length = 1552 bp; see Materials and methods). From left to right, the near-identical groups are named in **Supplementary file 3** as 156, 18, 117, 210, 22, 29, 44, 19, 184, 6, 208, 139, 32, 26, 10, 192, 217.

## Plasmid clustering reveals a diverse but intertwined population structure across niches

Near-identical plasmids shared across niches are a likely signature of recent transfer events, but we also wanted to examine the wider plasmid population structure. We therefore agnostically clustered all plasmids based on alignment-free sequence similarity (clusters were groups of n≥3 plasmids; see Materials and methods and *Appendix 1—figures 4 and 5*). We defined *n=247* plasmid clusters with median 5 members (IQR = 3–10, range = 3–123) recruiting 71% (2627/3697) of the plasmids. The remainder were either singletons (i.e. single, unconnected plasmids; 19% [718/3697]) or doubletons (i.e. pairs of connected plasmids; 10% [352/3697]). By bootstrapping *b*=1000 ACs for plasmid clusters, doubletons, and singletons found against number of isolates sampled (*Appendix 1—figure 6*; see Materials and methods), we estimated that the rarefaction curve had a Heap's parameter $\gamma$=0.75, suggesting further isolate sampling would likely detect more plasmid diversity and clusters.

Of the plasmid clusters, *n=69/247* (28%) had at least 10 members, representing 50% (1832/3697) of all plasmids (*Figure 3a*). 122/247 (49%) clusters contained BSI plasmids and plasmids from at least one other niche. This included 73/247 (30%) of clusters with both BSI and livestock-associated plasmids, representing *n=38* unique plasmid replicon haplotypes (i.e. combinations of replication proteins) of which only 24% (9/38) were Col-type plasmids, which are often well conserved and carry few genes (*Rozwandowicz et al., 2018*). 72/247 (29%) of clusters contained both BSI and influent/effluent/downstream plasmids, reflecting a route of Enterobacterales dissemination into waterways. In contrast, only 18/247 (7%) of clusters contained both BSI and upstream waterway plasmids, of which most (13/18 [72%]) also contained influent/effluent/downstream plasmids.

Overall, plasmid clusters scored high homogeneity (*h*) but low completeness (*c*) with respect to biological and ecological characteristics (non-putative PTUs [*h*=0.99, *c*=0.66]; replicon haplotype [*h*=0.92, *c*=0.69]; bacterial host sequence type (ST) [*h*=0.84, *c*=0.14] in *Figure 3b*; predicted mobility [*h*=0.93, *c*=0.20] in *Figure 3c*). This indicated that clustered plasmids often had similar characteristics, but the same characteristics were often observed in multiple clusters. When scoring plasmid clusters against broad sampling niche (BSI, livestock-associated, or WwTW-associated; *Figure 3a*), homogeneity was low (*h*=0.12, *c*=0.61), indicating mixed clusters. The imperfect homogeneity is to be anticipated as replicon haplotypes and mobilities can vary within plasmid families, and plasmid families can have diverse host ranges (*Redondo-Salvo et al., 2020*).

Plasmids carrying AMR genes were found in 21% (52/247) of the plasmid clusters (i.e. 'antimicrobial resistance gene [ARG]-carrying clusters'), representing *n=550* plasmids (*Figure 3d*). Of the ARG-carrying clusters, 92% (48/52) contained at least one beta-lactamase-carrying plasmid (*n=437* plasmids in total). AMR genes were present in a median 67% of ARG-carrying cluster members (IQR = 28–100%, range = 3–100%). This highlights that AMR genes are not necessarily widespread on genetically similar plasmids and can be potentially acquired multiple different times through the activity of smaller MGEs (e.g. transposons) or recombination. For example, cluster 12 was a group of *n=42* conjugative, PTU-$F_E$ plasmids found in BSI, wastewater, and waterways. Of these, 31% (13/42) carried the AMR gene *bla*$_{TEM-1}$, and in a range of genetic contexts: *n=9/13 bla*$_{TEM-1}$ genes were found within Tn*3* and *n=4/13* were carried without a transposase, of which *n=2/4* were found with the additional AMR genes *aph(6)-Id*, *aph(3")-Ib*, and *sul2*. F-type plasmids were the most common AMR gene carriers (61% [337/550] of all ARG-carrying cluster plasmids), underlining the known role of F-type plasmids in AMR gene dissemination (*Matlock et al., 2021a*).

The beta-lactamase *bla*$_{TEM-1}$ was the most common AMR gene detected (8% of total AMR gene annotations [424/5402]; see Materials and methods). In terms of sequence length (bp), plasmids made up 3.1% of the overall dataset but 13.8% of the *bla*$_{TEM-1}$-carrying proportion. Of the plasmid clusters, 16% (39/247) carried *bla*$_{TEM-1}$, and of these nine clusters were seen in human BSI and at least one other niche. Plasmid clusters either variably or always carrying *bla*$_{TEM-1}$ were strongly associated with BSI (p<0.01, Chi-squared test $X^2$=8.19, 33/161 of BSI clusters containing *bla*$_{TEM-1}$ vs. 5/86 for non-BSI

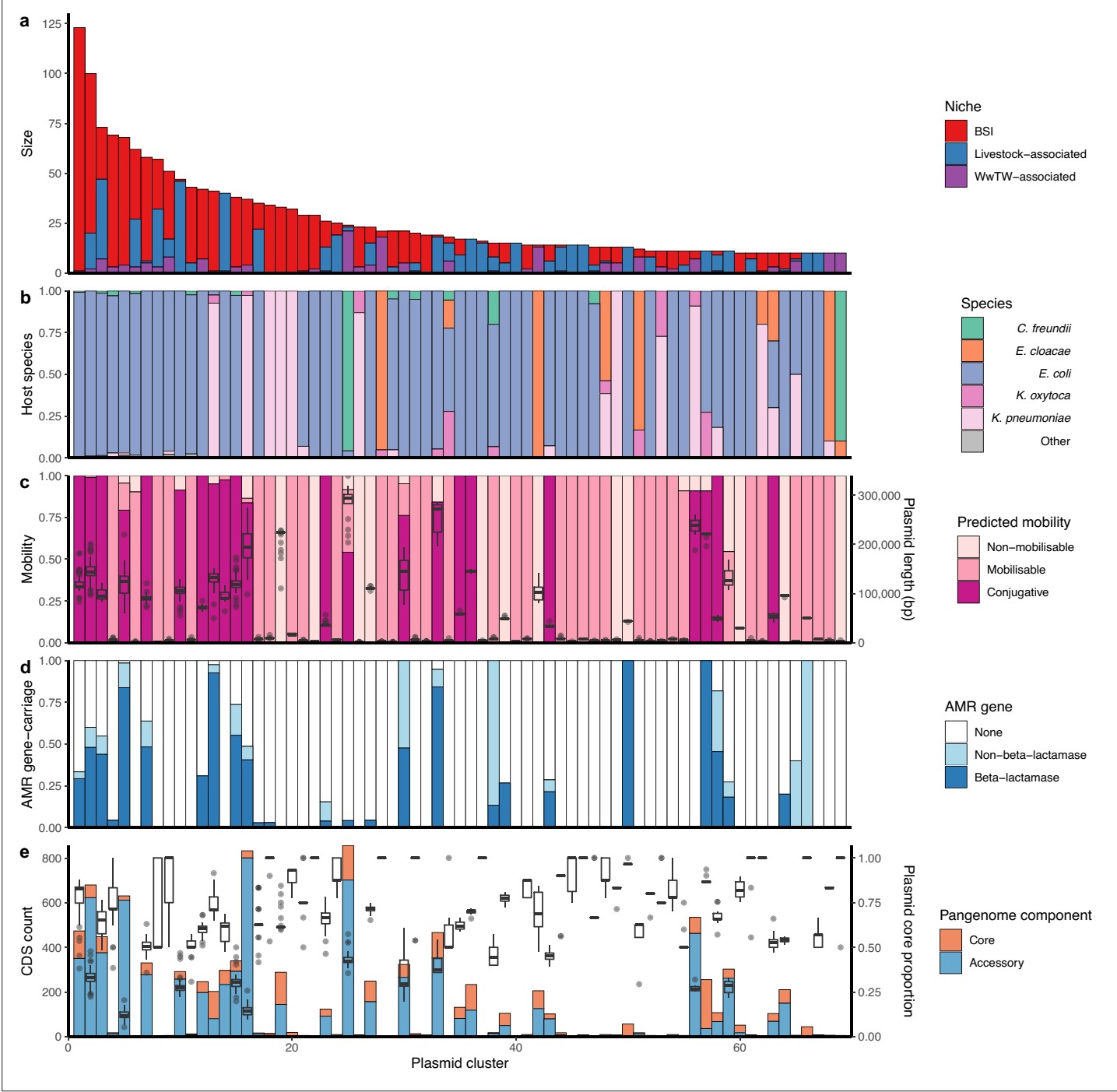

**Figure 3.** Genetically similar plasmids shared between niches. (**a**) Size of plasmid clusters with at least 10 members, coloured by niche. Size of clusters decreases from left to right. (**b**) Proportion of plasmid host species by cluster. (**c**) Plasmid mobility class and size: Left-hand axis shows proportion of plasmids with a predicted mobility class by cluster. Right-hand axis shows plasmid length boxplots by cluster. (**d**) Proportions of antimicrobial resistance (AMR) gene carriage by cluster. (**e**) Plasmid core and accessory genomes: Left-hand axis shows the count of core and accessory coding sequences (CDS) for the entire cluster as a bar chart. Right-hand axis shows plasmid core-gene proportions (i.e. core CDS/total CDS for each plasmid) as a boxplot.

clusters) and carried a higher number of other AMR genes (p<0.01, Wilcoxon text of $bla_{TEM-1}$-plasmid clusters vs. others; see *Appendix 1—figure 7*).

## An intertwined ecology of plasmids across human and livestock-associated niches

Plasmids can change their genetic content, particularly when subject to new selective pressures (*Rodríguez-Beltrán et al., 2021*; *Pesesky et al., 2019*). Many plasmids have a structure with a 'backbone' of conserved core genes and a 'cargo' of variable accessory genes (*Orlek et al., 2017b*; *Matlock et al., 2021a*; *Coluzzi et al., 2022*). We wanted to explore evidence for cross-niche plasmids with minimal mutational evolution in a shared backbone (compatible with approximately years of evolutionary separation) but variable accessory gene repertoires.

We first conducted a pangenome-style analysis (see Materials and methods) on the *n=69/247* plasmid clusters with at least 10 members. For each cluster, we determined 'core' (genes found in ≥95% of plasmids) and 'accessory' gene repertoires (found in <95% of plasmids). Within clusters, we found median 9 core genes (IQR = 4–53, range = 0–219), and median 9 accessory genes (IQR = 3–145, range = 0–801) (*Figure 3e*). Core genes comprised a median proportion 42.2% of the total pangenome sizes (IQR = 20.9–66.7%). At an individual plasmid level, core genes shared by a cluster comprised a median 62.5% of each plasmid's gene repertoire (IQR = 37.4–83.3%; *Figure 3e*). Putatively conjugative plasmids carried a significantly higher proportion of accessory genes in their repertoires than mobilisable/non-mobilisable plasmids (Kruskal-Wallis test followed by Dunn's test [$H(2)=193.01$, p-value <0.001]).

Using multiple sequence alignments of the core genes within each cluster, we produced maximum likelihood phylogenies (see *Supplementary file 1* and Materials and methods). For this step, we only considered the *n=62/69* clusters where each plasmid had ≥1 core gene. With the *n=27/62* clusters that contained both BSI and livestock-associated plasmids, we measured the phylogenetic signal for plasmid sampling niche using Fritz and Purvis' *D* (see *Supplementary file 2* and Materials and methods). The analysis indicated that the evolutionary history of plasmid clusters is neither strictly segregated by sampling niche nor completely intermixed, but something intermediate.

Alongside the core-gene phylogenies, we generated gene repertoire heatmaps (example cluster 2 in *Figure 4a–b*; all clusters and heatmaps in *Supplementary file 1*). By visualising the genes in a consensus synteny order (see Materials and methods), the putative backbone within each plasmid cluster is shown alongside its accessory gene and transposase repertoire. This highlights how plasmids might gain/lose accessory functions within a persistent backbone. Log-transformed linear regression revealed a significant relationship between Jaccard distance of accessory genes presence against core-gene cophenetic distance ($y=0.080\log(x)+0.978$, $R^2=0.47$, $F(1,52988)=4.75e4$, p-value <0.001; see *Appendix 1—figure 8* and Materials and methods).

## Plasmid dissemination between human and livestock-associated niches is not structured by bacterial host

Alongside vertical inheritance, conjugative and mobilisable plasmids are capable of inter-host transfer, crossing between bacterial lineages, species, up to phyla (*Redondo-Salvo et al., 2020*). Phylogenetic analysis can determine whether plasmid evolution between BSI and livestock-associated niches is driven by host clonal expansion or other means, as well as allow us to explore the early emergence of AMR gene carrying plasmids.

As a detailed example, we evaluated the largest plasmid cluster containing both human and livestock-associated plasmids (cluster 2, *n=100* members). All plasmids carried at least one F-type replicon and were all putatively conjugative, with 75% (75/100) and 25% (25/100) assigned PTU-$F_E$ and a putative PTU, respectively. Further, 48% (48/100) plasmids carried $bla_{TEM-1}$, and 51% (51/100) carried more than one AMR gene. All host chromosomes were *E. coli* except OX-BSI-481_2 (*S. enterica* ST 2998; hereon omitted from the analysis). The *n=99 E. coli* isolates represented six phylogroups: A (5/99), B1 (18/99), B2 (52/99), C (14/99), D (7/99), and G (3/99; see Materials and methods).

*Figure 4b–c* shows the plasmid core-gene phylogeny ($T_{plasmid}$) and the *E. coli* host core-gene phylogeny ($T_{chromosome}$). The *E. coli* phylogeny was structured by six clades corresponding to the six phylogroups (see Materials and methods). We found low congruence between the plasmid core-gene phylogeny and the chromosomal core-gene phylogeny as seen in the central 'tanglegram' (i.e. lines connecting pairs of plasmid and chromosome tips from the same isolate). Additionally, we calculated

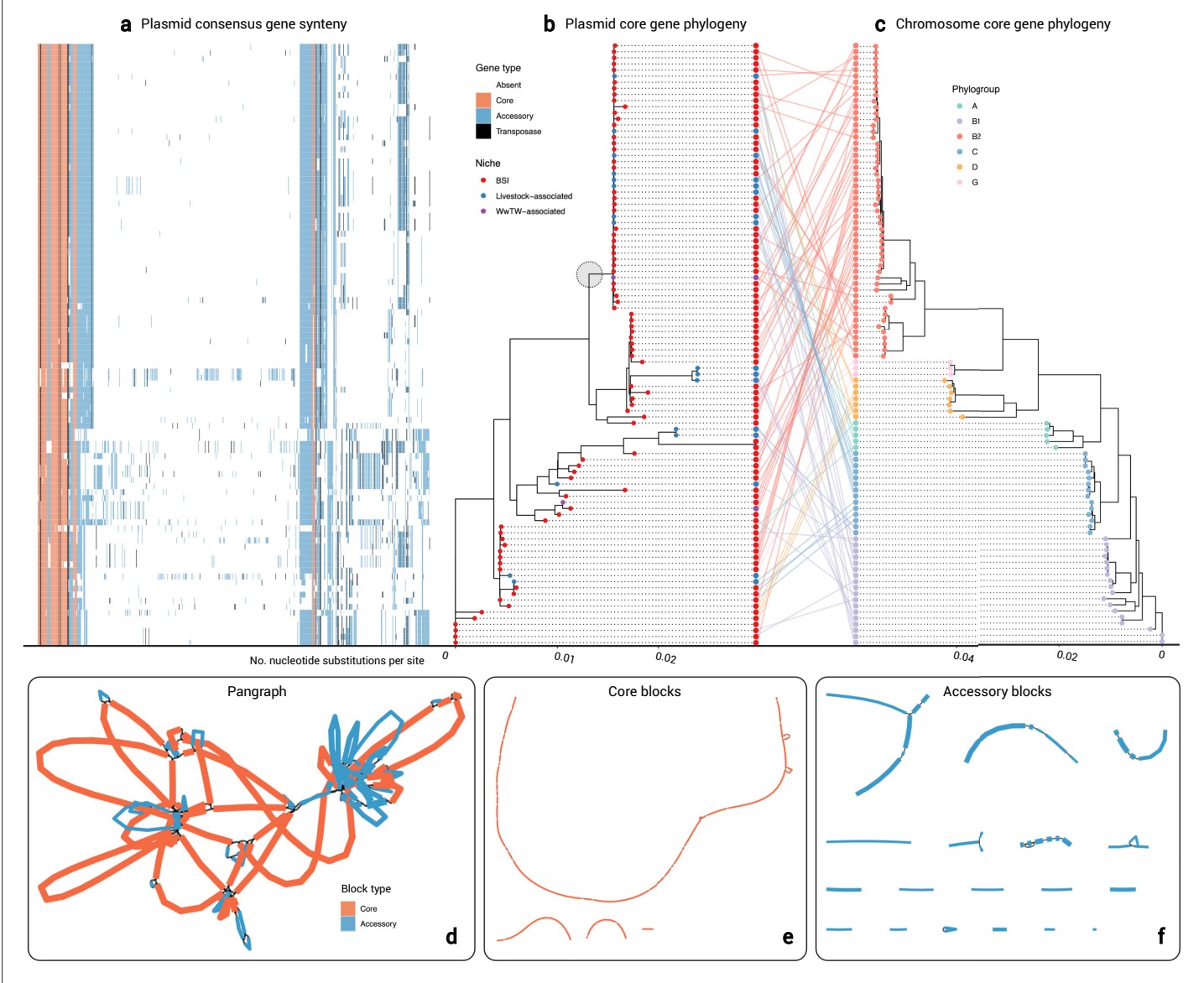

**Figure 4.** Cluster 2 plasmid and host evolution. (**a**) Consensus gene ordering for plasmid cluster 2, coloured by gene type (total *n=99* plasmids; *n=1* *Salmonella enterica* isolate omitted). Genes are coloured by core, accessory, or transposase. (**b**) Plasmid core-gene phylogeny with tips coloured by sampling niche. The grey circle highlights the clade of *n=44* plasmids which were further analysed. (**c**) Plasmid host chromosome core-gene phylogeny with tips coloured by sampling niche. Plasmid and host phylogeny tips are connected in a 'tanglegram' which connects pairs of plasmids and chromosomes from the same isolate. (**d**) Visualisation of the pangraph for *n=44* plasmids in the grey-circled clade in (**b**). Blocks are coloured by presence in plasmids. (**e**) Core blocks (found in at least 95% of the *n=44* plasmids). (**f**) Accessory blocks (found in less than 95% of the *n=44* plasmids).

a Robinson-Foulds distance $RF(T_{\text{plasmid}}, T_{\text{chromosome}})=162$, reflecting a high number of structural differences between the phylogenies (see Materials and methods). There was some evidence of plasmid structuring by niche (Fritz and Purvis' $D=0.24$; see Materials and methods).

Within the plasmid phylogeny, there was a clade of *n=44* plasmids (support 100%; circled in grey in **Figure 4b**) containing both BSI and livestock-associated plasmids, which were within median 4 core-gene SNPs of each other (IQR = 2–8, range = 0–59). Estimating plasmid evolution at an approximate rate of one SNP per year (see Materials and methods) would give a median time to most recent common ancestor of the backbone at approximately 4 years prior to sampling, consistent with recent movement between human and livestock-associated niches. This plasmid clade was mainly present in phylogroup B2 (20/44), but also A (3/44), B1 (9/44), C (8/44), and D (4/44), suggesting plasmid movement. Further, 77% (34/44) of plasmids within the clade carried *bla*~TEM-1~ (BSI: 25/34,

livestock-associated: 8/34, WwTW-associated: 1/34), and 82% (36/44) carried ≥1 AMR gene, high-lighting the role of plasmids in cross-niche dissemination of AMR.

To examine the evolution of entire plasmid sequences within the clade, we represented all *n=44* plasmids as a 'pangraph' (*Figure 4d*; see Materials and methods). Briefly, pangraph converts input sequences into a consensus graph, where each sequence is a path along a set of homologous sequence alignments, i.e., 'blocks', which in series form 'pancontigs'. Filtering for 'core blocks' (i.e. those found in ≥95% plasmids), we found 4 pancontigs (40 blocks total), with the longest 98,269 bp (total length 125,369 bp), indicating a putative plasmid backbone (*Figure 4e*). Then, filtering for 'accessory blocks' (i.e. those found in <95% plasmids), we found 18 pancontigs (39 blocks total), with median length 2380 bp (total length 63,753 bp), forming the accessory gene repertoire (*Figure 4f*). Core and accessory pancontigs contained 22% (57/261) and 78% (204/261) of gene annotations, respectively, of which over half encoded hypothetical proteins (51%; 134/261; see *Supplementary file 4* and Materials and methods). Core annotations included replication (*repB*) and conjugation (*finO*, *traI*, *traM*) proteins, whereas accessory gene annotations included AMR (*bcr*, *blaTEM*, *tetA*, *tetR*) and mercury resistance (*merA*, *merC*, *merP*, *merT*) proteins. Transposase/insertion sequence annotations were disproportionately found in accessory pancontigs (88%; 38/43) versus core pancontigs (12%; 5/43). This points to a persistent plasmid backbone structure with loss/gain events at particular 'hotspots' as well as rearrangements.

## Discussion

Sharing of plasmids between different niches is normally focused on those carrying AMR genes that are of particular current clinical concern, such as extended-spectrum beta-lactamase (ESBL) or carbapenemase genes, meaning we lack information on the vast 'denominator' of background plasmid sharing, and on the dissemination of other AMR genes which are now widespread in clinical isolates and from which important insights might be gained. By analysing a dataset of *n=3697* systematically collected Enterobacterales plasmids sampled from human BSI, livestock- and WwTW-associated sources in a geographically and temporally restricted context, we found evidence supporting significant plasmid dissemination across niches, putting those which carry AMR genes of current major clinical concern into context. We found 225 instances of shared, near-identical plasmid groups, 25% of which were found across multiple bacterial STs, 4% across multiple bacterial species, and 8% in both human BSI and ≥1 non-BSI niche. Beyond this near-identical sharing, we analysed 'clusters' of plasmids and found that 73/247 clusters contained plasmids seen in both human BSIs and other contexts. Approximately a fifth (52/247) of plasmid clusters contained plasmids carrying AMR genes (*n=550* plasmids). Our results suggest the need for broad, unselected, and detailed sampling frames to fully understand plasmid diversity and evolution, and to evaluate the 'One Health' risk of AMR associated with plasmid sharing across niches.

Whilst many plasmid clusters were strongly structured by host phylogeny and isolate source, some plasmids from human BSIs were highly genetically related to those in other niches, including livestock. However, not all of these carried AMR genes. Our results highlight the potential routes for transfer that exist through similar plasmids. However, recovering these instances of putative sharing is a sampling challenge. Accumulation curve analyses suggested increasing the size of our dataset would have led to further near-identical matches at an approximately linear rate, meaning even a dataset of this size captures only a small fraction of the true extent of plasmid sharing between human clinical and other non-human/clinical niches. This presents a challenge for designing appropriately powered studies. Had we only sampled *n=100* livestock-associated isolates (i.e. around 20% of our actual sample), there was only a 39% chance that we would have detected ≥5 matches with BSI plasmids (*Appendix 1—figure 9*).

Understanding the evolutionary history, distribution, and epidemiology of well-known genes in environmental plasmids may offer insights into the future trajectories of more recently emerged genes. For example, the first plasmid-encoded beta-lactamase to be described was $bla_{TEM-1}$, identified in 1965 in an *E. coli* isolate in Greece (*Datta and Kontomichalou, 1965*) and now widely prevalent in Enterobacterales (*Bush and Bradford, 2020*). $bla_{TEM-1}$ has a narrow spectrum of activity and is now less clinically concerning than newer genes which mediate broad-spectrum resistance, but in our dataset $bla_{TEM-1}$ was strongly associated with plasmid clusters seen in BSI and with the carriage of other AMR genes. $bla_{TEM-1}$ may continue to play an important role in the spread of AMR-carrying plasmids

which can transfer recently emerged genes, and similarities in its association with plasmids and other smaller transposable MGEs may reflect the future trajectory of other AMR genes of more recent clinical concern such as ESBLs and carbapenemases.

Given that plasmids observed in BSI isolates represent small proportion of human Enterobacterales diversity, many more sharing events may occur in the human gut (*Forster et al., 2019*) which we only sampled incompletely using wastewater influent as a proxy. The human colon contains around $10^{14}$ bacteria (*Sender et al., 2016*), with large ranges of *Enterobacteriaceae* abundance. Further, even small numbers of across-niche sharing events, such as transfer events of important AMR genes from species-to-species or niche-to-niche, may have significant clinical implications, as has been seen with several important AMR genes globally. Future studies need to carefully consider the limitations of sampling frames in detecting any genetic overlap, given both substantial diversity and the effects of niches and geography (*Shaw et al., 2021*; *Hanage, 2019*).

By examining plasmid relatedness compared to bacterial host relatedness in *E. coli*, we demonstrated that plasmids seen across different niches are not necessarily associated with clonal lineages. Using a pangenome-style analysis, we showed that plasmids can share sets of near-identical core genes alongside diverse accessory gene repertoires. While plasmids with more distantly related core genes tended to have dissimilar accessory gene content, plasmids with more closely related core genes shared a wide range of accessory gene content. This would be consistent with a hypothesis of persistent 'backbone' structures gaining and losing accessory functions as they move between hosts and niches. We suggest that this mode of transfer might be worth considering. Evolutionary models for plasmids which can accommodate well-conserved backbone evolution alongside accessory structural changes and gain/loss events are urgently needed. Estimating plasmid evolutionary rates remains a challenge, with little known about appropriate values for mutation rates in plasmids, and even less for non-mutational processes such as gene gain/loss.

Our study had several limitations. Our non-BSI isolates were not as temporally varied as the BSI isolates, meaning we could not fully explore temporal evolution. Although we evaluated four bacterial genera, 72% (1044/1458) of our sequenced isolates were *E. coli*, and so our analyses and findings are particularly focused on this species. Additionally, we did not sample livestock-associated niches densely enough to explore individual livestock types (cattle/pig/poultry/sheep) sharing plasmids with BSI isolates (see *Appendix 1—figure 9*). Isolate-based methodologies are limited in evaluating the true diversity of the niches sampled; composite approaches including metagenomics might shed additional insight in future studies. Further, the exact source of an isolate is poorly defined for wastewater/waterway isolates as they act as a confluence of multiple sources, although they represent important niches in their own right. We only analysed plasmids from complete genomes, i.e., where the chromosome and all plasmids were circularised, meaning we disregarded ~23% and ~33% of BSI and non-BSI assemblies, respectively. The exclusive use of complete assemblies was to ensure full plasmid sequences could be examined in their full genomic context. We only focused on plasmids as horizontally transmissible elements here; detailed study of other smaller MGEs across-niches would represent interesting future work. We have also investigated a limited subset of Enterobacterales: plasmid sharing likely extends to other bacterial hosts not investigated here. Lastly, our isolate culture methods for livestock-associated samples may not have been as sensitive for the identification of *Klebsiella* spp. as for other Enterobacterales such as *Escherichia*, as we did not use enrichment and selective culture on Simmons citrate agar with inositol (*Rodrigues et al., 2022*). This may have limited our ability to study the epidemiology of livestock *Klebsiella* plasmids.

In conclusion, this study presents to our knowledge the largest evaluation of systematically collected Enterobacterales plasmids across human and non-human niches within a geographically and temporally restricted context. Near-identical plasmids can be found in different niches, pointing to putative dissemination, although this dynamic likely varies by plasmid cluster; the proportion of near-identical plasmid groups that were found across niches was 8% (17/225) and influenced by sample size. We demonstrate a likely intertwined ecology of plasmids across human and non-human niches, where different plasmid clusters are variably but incompletely structured 1475 and putative 'backbone' plasmid structures can rapidly gain and lose accessory genes following cross-niche spread. Future 'One Health' studies require dense and unselected sampling, and complete/near-complete plasmid reconstruction, to appropriately understand plasmid epidemiology across niches.

## Materials and methods

Livestock-associated isolates *n=247* Enterobacterales isolates from farm-proximate soils and poultry faeces (*n=19* farms; *n=5* cattle, *n=4* pig, *n=5* poultry, *n=5* sheep) were collected and sequenced for this study in 2017–2020. DNA extraction and sequencing was performed as in *Shaw et al., 2021*. Genomes were hybrid assemblies reconstructed using Unicycler (*Wick et al., 2017*) (v. 0.4.4; default hybrid assembly parameters except `--min_component_size 500 --min_dead_end_size 500`). Only complete assemblies (plasmids and chromosomes) were considered (*n=162/247*).

### BSI isolates

Sequenced Human BSI Enterobacterales isolates from patients presenting to *n=4* hospitals within Oxfordshire, UK, September 2008–December 2018, as described in *Lipworth et al., 2021*, were also included. Although all patients were sampled in Oxfordshire, a total of *n=505/738* patients resided in Oxfordshire, *n=133/738* in surrounding counties, and *n=100/738* had location information omitted. Only complete assemblies (*n=738/953* total assembled) were considered.

### Other livestock-associated and WwTW-associated isolates

Enterobacterales isolates from faeces from the *n=14* non-poultry farms and wastewater influent, effluent, and waterways upstream/downstream of effluent outlets surrounding *n=5* WwTWs, across three seasonal timepoints in 2017 were included (as in *Shaw et al., 2021*), were included. Only complete assemblies (*n=558/827* total assembled) were considered.

### Taxonomic assignment

Chromosome STs were determined with *mlst, 2017* (v. 2.19.0; PubMLST database; *Jolley and Maiden, 2010*). For the *n=11/1458* chromosomes which could not be typed with mlst, species were determined with the PubMLST 'species ID' web-tool (*Jolley et al., 2018*), for which all had a support = 100, except for *Lelliottia nimipressuralis* (support = 83). Of these, *n=5/11* were from BSI, *n=4/11* from livestock, and *n=2/11* from effluent/downstream of WwTWs. From the BSI isolates, we also included *n=2 Aeromonas* spp., a non-Enterobacterales genus from the wider *Gammaproteobacteria* class.

### Chromosome trees

Trees for *E. coli* and *K. pnemoniae* chromosomes were produced using Mashtree (*Katz et al., 2019*) on 'accurate' mode (`--mindepth 0 --numcpus 12`).

### PTU classification

Plasmids were assigned a PTU using COPLA (*Redondo-Salvo et al., 2021*) (default parameters except -t circular, -k Bacteria, -p Pseudomonadota, -c Gammaproteobacteria, and -o Enterobacterales) (*Redondo-Salvo et al., 2020*). COPLA compares query plasmids to a database of PTU reference plasmids, assigning a PTU when both (i) the ANI >0.7 along 50% of the length of the smallest plasmid in the comparison and (ii) a graph-neighbouring condition to existing PTU clusters is satisfied. The COPLA reference database contains over 10,000 curated, non-redundant plasmids retrieved from the 84th NCBI RefSeq database in 2017 (*Pruitt et al., 2007*). We contextualised our plasmids within known plasmid diversity using COPLA to determine each plasmid's 'PTU' (see Materials and methods), which is designed to be equivalent to a 'species' concept for plasmids (*Redondo-Salvo et al., 2021*). Briefly, COPLA classifies query plasmids based on average nucleotide identity (ANI) against a non-redundant reference plasmid database where most plasmids have been assigned to a reference PTU (*Pruitt et al., 2007*). Within our sample, 64% (2369/3697) plasmids were assigned a PTU and 4% (135/3,697) a putative PTU (i.e. the query plasmid was clustered with three unclassified reference plasmids). This is consistent with a previous COPLA analysis of 1000 Enterobacterales plasmids which found that 63% were classified into a PTU (*Redondo-Salvo et al., 2021*). The remaining 32% (1193/3697) of plasmids were unclassified (i.e. connected set with less than four plasmids) highlighting the previously unsampled plasmid diversity within our dataset. In total, we found *n=67* known PTUs, containing a median 9 plasmids (IQR = 4–30, range = 1–556), where the largest assigned PTU (556/2504) was PTU-F$_E$, corresponding to F-type *Escherichia* plasmids (*Matlock et al., 2021a*; *Rozwandowicz et al., 2018*). The proportion of unclassified plasmids was higher in environmental/livestock samples (33%; 385/1155)

versus BSI samples (26%; 485/1880), emphasising the underrepresentation of non-human plasmids in reference plasmid databases.

## Plasmid annotation

All plasmids were annotated with Prokka (*Seemann, 2014*) (v. 1.14.5) with default parameters. For replicon typing, Abricate (*Seeman, 2015*) (v. 1.0.0) was used with the PlasmidFinder (*Carattoli et al., 2014*), ISfinder (*Siguier et al., 2006*), and BacMet (*Pal et al., 2014*) databases with default parameters and output filtered for 80% minimum coverage. For annotating AMR genes, NCBI Antimicrobial Resistance Gene Finder (AMRFinderPlus) (*Feldgarden et al., 2021*) (v. 3.10.18) was used with default parameters. To assign putative plasmid mobilities, we used MOB-typer from MOB-suite (*Robertson and Nash, 2018*) (v. 3.03) with default parameters. MOB-typer predicts mobility based on annotations of relaxase (*mob*), mating pair formation (MPF) complex, and *oriT* genes. Briefly, a plasmid is putatively labelled conjugative if it has both relaxase and MPF, mobilisable if it has either relaxase or *oriT* but no MPF, and non-mobilisable if it has no relaxase and *oriT*.

## Near-identical plasmid screening

Groups of near-identical plasmids were detected as connected components in a plasmid-plasmid network with Mash distance (*Ondov et al., 2016*) (v. 2.3; default parameters except sketch size -s 1000000) weighted edges, at a threshold $d<0.0001$. Briefly, Mash distance estimates an evolutionary distance on a reduced-length MinHash sketch of the sequences. Since Mash is a probabilistic estimate of evolutionary distance, we confirmed the probability of seeing any of our pairwise Mash distances in the near-identical groups by chance was 0. For whole genomes, Mash distance has a strong positive correlation with ANI (*Figueras et al., 2014*). We also required the shortest plasmid to be within 1% length (bp) of the longest plasmid, to account for assembly errors. Network analysis was performed using the igraph (*Csardi and Nepusz, 2006*) library (v. 1.2.7) in R.

The stringency of a *k*-mer-based distance threshold for near-identical plasmid clustering is equivalent to a threshold on the Jaccard index (i.e. rearranging the Mash distance calculation ($d = \frac{-1}{k} \ln \left( \frac{2j}{1+j} \right)$) with $d=10^{-4}$ and $k=21$ gives a Jaccard index threshold of $j=0.9958$). The effect of this threshold varies with plasmid size: at very small plasmid sizes, clusters contain only identical plasmids because the presence of a single SNP means plasmids are placed in different clusters. For example, two 1552 bp plasmids with a single SNP (e.g. RHB03-C05_6 and RHB02-C22_6) will have a Mash distance of $d=5.0 \times 10^{-4}$ ($>10^{-4}$ threshold). In contrast, at length = 150 kb a single SNP (not at the start/end of the plasmid) would lead to $d=5.6 \times 10^{-6}$ ($<<10^{-4}$ threshold); even two 150 kbp plasmids with ~30 SNPs would have $d\approx2\times10^{-4}$ ($>10^{-4}$ threshold) and so be split into near-identical plasmids. Our analysis of plasmid sharing is therefore maximally conservative at small plasmid sizes but remains highly conservative for large plasmids.

## Accumulation and rarefaction curves

To generate an accumulation curve, isolates were sampled without replacement in a random order. For each isolate, the new plasmid diversity was recorded. For *Appendix 1—figure 3*, we recorded the number of new near-identical plasmid groups and singletons. For *Appendix 1—figure 9*, we recorded the number of near-identical matches with BSI plasmids from only environmental/livestock isolates. For *Appendix 1—figure 6*, we recorded the number of new clusters, doubletons, and singletons. A bootstrapped average of $b=1000$ accumulation curves was plotted for the rarefaction curve. The bootstraps were also used to estimate Heap's parameter (γ) by fitting a linear regression to log-log transformed data using standard R libraries. For $\gamma<0$, it is possible to sample the entire diversity, and for $1>\gamma>0$, the diversity will increase with every additional sample (*Tettelin et al., 2008*).

## Plasmid similarity

Plasmid Jaccard index (*JI*) was calculated using Mash (*Ondov et al., 2016*) (v. 2.3; default parameters except sketch size `-s 1000000`). The *JI*, given by

$$JI\left(A, B\right) = \frac{|A \cup B|}{|A \cap B|}$$

where *A*, *B* are the sets of *k*-mers of plasmids *a*, *b*, respectively. This measures extent of *k*-mer sharing between plasmids, range = 0–1, where 1 indicates an identical *k*-mer repertoire. Since the sketch size was larger than the plasmid lengths (except for one plasmid in the dataset, OX-ENV-67_2, which was larger than 1 Mbp at 1,310,597 bp and was not clustered; the next smallest was OX-WTW-80_2 at 394,284 bp), the calculated Jaccard indices were almost always exact.

## Plasmid network and clustering

The determination of the plasmid-plasmid network, threshold, and clusters could be achieved with several alternative methodologies. Plasmid networks have previously been constructed by full sequence alignments (*Yamashita et al., 2014*), annotated genes (*Branger et al., 2018*), and alignment-free Mash distances (*Matlock et al., 2021a*; *Acman et al., 2020*; *Jesus et al., 2019*). We chose to use the Jaccard index of entire plasmid *21*-mer distributions to capture coding sequences, their immediate contexts (*Matlock et al., 2021b*; *Arcilla et al., 2016*), and intergenic regions (*Zhi et al., 2015*; *Delihas, 2009*), all of which have known importance to bacterial evolution. Further, our contained previously unsampled diversity as seen by the PTU analysis, and because reference-based classifications such as MOB and replicon typing schemes are known to be incongruent (*Orlek et al., 2017a*) or unreliable: 16% (602/3697) of our plasmids had an unidentifiable replicon type, which is not uncommon (*Rozwandowicz et al., 2018*). The evolutionary histories of plasmids can incorporate multiple gain, loss, and rearrangement events in addition to mutations (*Kizny Gordon et al., 2020*), and as such, traditional measures of genetic relatedness (e.g. single nucleotide variant thresholds) used for genomic epidemiology of whole genomes are likely less appropriate here. These similarities formed the edge weights in a plasmid-plasmid network, which was subsequently thresholded to sparsify the network and allow the detection of clusters.

Network thresholding to some extent depends subjectively on the dataset, with trade-offs between successfully revealing the underlying structure of plasmid relationships without excessively separating relatives. We chose a data-driven threshold as adopted by *Branger et al., 2018*, for their plasmid network, which examined the evolution of connected components within the network. This ensured the threshold was chosen where the regime of connected component evolution approximately stabilises, minimising excessive network breakup. The threshold was chosen at *JI* = 0.5, meaning that edges between plasmids with *JI* <0.5 were deleted from the network. From this threshold onwards, both the number of connected components and the number of singletons steadily increased at a similar rate (*Appendix 1—figure 4*). This regime indicates an approximately stable non-singleton structure from *JI* = 0.5 onwards.

We defined plasmid clusters as groups of n≥3 plasmids with high within-cluster similarities and low between-cluster similarities. Plasmid clusters were detected using the Louvain algorithm which optimises the network modularity by iterative expectation maximisation (*Blondel et al., 2008*). This aims to maximise the density of edges within clusters against edges between clusters. Though non-deterministic, the Louvain algorithm showed low variation in cluster distribution over 50 runs, consistent with reproducible segregation of plasmids in clusters (range of clusters detected: 245–247; *Appendix 1—figure 5*). The algorithm was implemented using the Python-Louvain (v. 0.16) Python module. Although the algorithm is non-deterministic, multiple runs demonstrated minimal variation at our chosen network threshold. Overall, these approaches add to the growing literature describing suitable methodologies for clustering plasmids.

Near-identical plasmid groups were also included in the wider cluster analysis, as many were cross-compartmental and found across bacterial hosts (see earlier, *Figure 2*). Of the *n=194/225* groups which were clustered, 100% (194/194) had all members fall within the same plasmid cluster, with *n=30/247* clusters containing multiple near-identical plasmid groups. Only 6% (14/247) of plasmid clusters comprised exclusively near-identical plasmid groups, suggesting that near-identical groups of plasmids often have nearby genetically related plasmids. Examining the entire PTU distribution within clusters, most contained at least one unclassified plasmid (51%; 127/247) or plasmid assigned a putative PTU (9%; 23/247). However, many clusters exclusively contained just one known PTU (42%; 105/247).

## Cluster homogeneity and completeness

Homogeneity (*h*) and completeness (*c*) are dual conditional entropy-based measures, independent of cluster and metadata label distributions (*Rosenberg and Hirschberg, 2007*). A clustering satisfies

homogeneity ($h=1$) if all cluster members have the same metadata label type. Consider a network with $N$ nodes, partitioned by a set of metadata labels, $M = \{m_i | i = 1, \ldots, n\}$, and a set of communities, $C = \{c_j | j = 1, \ldots, m\}$. Let $A = \{a_{ij}\}$ represent the $ij^{\text{th}}$ entry in the contingency table of partitions. Hence, $a_{ij}$ counts the number of nodes with label $m_i$ in community $c_j$. We then say

$$h = \begin{cases} 1 & \text{if } H(M, C) = 0 \\ 1 - \dfrac{H(M|C)}{H(M)} & \text{else} \end{cases}$$

where

$$H(M|C) = -\sum_{c=1}^{|C|} \sum_{m=1}^{|M|} \frac{a_{mc}}{N} \log \frac{a_{mc}}{\sum_{c=1}^{|M|} a_{mc}}$$

and

$$H(M) = -\sum_{m=1}^{|M|} \frac{\sum_{c=1}^{|C|} a_{mc}}{n} \log \frac{\sum_{c=1}^{|C|} a_{mc}}{n}$$

are the conditional entropy of the metadata given the clusters and the entropy of the clusters, respectively $H(M|C) = 0$ when the cluster partition coincides with the metadata partition, and no new information is added. A cluster partition satisfies completeness ($c=1$) if all instances of a metadata label type are assigned the same cluster. Completeness is defined dually by

$$c = \begin{cases} 1 & \text{if } H(C, M) = 0 \\ 1 - \dfrac{H(C|M)}{H(C)} & \text{else} \end{cases}$$

The measures were calculated using the clver library (v. 0.1.1) in R.

## Cluster pangenome analysis

Cluster pangenomes were generated using Panaroo (***Tonkin-Hill et al., 2020***) (v. 1.2.9) with parameters default except `--clean-mode sensitive --aligner mafft -a core --core_threshold 0.95`. For core-gene alignments, the threshold was set at minimum 95% presence amongst clustered plasmids, whereby they were aligned using MAFFT (***Katoh and Standley, 2013***) (v. 7.407) with default parameters. An identical approach was taken for the host chromosome phylogeny in ***Figure 4***. The median length of plasmids within a cluster was positively correlated with number of core genes ($R=0.85$, $t=13.4$, p-value $<2.2e-16$) and total pangenome size ($R=0.87$, $t=14.6$, p-value $<2.2e-16$).

## Plasmid core-gene phylogenies

Maximum likelihood core-gene phylogenies were generated using IQ-Tree (***Minh et al., 2020***) (v. 2.0.6) with parameters `-m GTR +F + I + G4 -keep-ident -T 2 -B 1000`. The substitution model used was general time reversible (GTR) using empirical base frequencies form the alignment (F), allowing for invariable sites (I) and variable rates of substitution (G4). We used $n=1000$ ultrafast bootstraps (B 1000; see ***Minh et al., 2013***) to visually inspect larger clades for support. Briefly, 95% support approximates a 95% probability that the clade is genuine. Only the $n=62/69$ clusters (excluding 6, 8, 26,2 9,3 2, 40, and 65) where every plasmid carried at least 1 core gene were analysed. Phylogenies were primarily plotted using the R library ggtree (***Yu et al., 2017***).

## Fritz and Purvis' *D*

Fritz and Purvis' $D$ measures phylogenetic signal for binary traits (***Fritz and Purvis, 2010***). First, we calculate the character state changes required to observe our phylogeny ($d_{obs}$). To account for phylogeny size and prevalence, $d_{obs}$ is standardised under the two null models: (i) tip labels are random permuted ($d_r$), and (ii) tip labels are distributed under the expectation of a Brownian motion model of evolution ($d_b$). Then, we define

$$D = (d_{obs} - \overline{d_b})/(\overline{d_r} - \overline{d_b}).$$

Hence, for $D{\approx}1$, $d_{obs}$ follows $d_r$ more closely, and for $D{\approx}0$, $d_{obs}$ follows $d_b$ more closely. We calculated $d_{obs}$ $n{=}10{,}000$ times and averaged the result, as well as calculate p-values for significant deviation from $d_r$ or $d_b$. $D$ was implemented using the R library caper (*Orme, 2013*). Fritz and Purvis' $D$ is normally used for cross-species analysis so is not benchmarked for plasmids. Results for phylogenies with less than 25 tips should be viewed more conservatively due to reduced statistical power in these instances.

We considered the binary 'trait' of human or livestock-associated isolate and estimated $D$ with $n{=}10{,}000$ permutations. We found 42% (11/26) clusters had $D{>}0.5$ (see *Supplementary file 2*). However, only 23% (6/26) of phylogenies were significantly different (p-value <0.05) from the conserved null model, compared to 50% (13/26) significantly different from the random null model.

## Consensus gene synteny heatmaps

For each cluster, we first generated a list of every possible pair of genes in the pangenome. Then for each plasmid, we counted the distance between these pairs, modulo the number of genes in the plasmid. If a gene was absent in a plasmid, NA was used. We then calculated the median of these values across all plasmids in the cluster. We then built a dendrogram from a hierarchical clustering of the median distances. The order of the tip labels in the dendrogram were then used as the 'consensus gene synteny'.

## Accessory gene distances

Plasmid accessory gene distances were calculated using pairwise Jaccard distances on gene presence-absences matrices. For plotting the cluster-wise plasmid core-gene cophenetic distance against accessory gene presence-absence Jaccard distance, only the $n{=}26/62$ clusters with at least 50 accessory genes were plotted. The log-transformed linear regression of Jaccard distance of accessory genes presence against core-gene cophenetic distance was fitted in R with standard libraries.

## Chromosome core-gene phylogeny

An identical approach was taken to the plasmid phylogenies. *E. coli* phylogroups were typed using EzClermont (*Waters et al., 2020*) (v. 0.7.0) with default parameters. Robinson-Foulds distance was calculated using the R library phangorn (*Schliep, 2011*).

## Plasmid mutation rate

Mutation rates per base pair in microbes typically arise from DNA replication and tend to be below $m{=}10^{-9}$ per site per generation (*Drake, 1991*) or perhaps as low as $10^{-10}$ per site per generation (*Foster et al., 2015*; *Wielgoss et al., 2013*). For a plasmid of size $L$, one therefore expects $L \times m$ mutations per plasmid per generation. For example, if the plasmid has $L{=}10^5$ then in each generation 1 in 10,000 plasmids will gain a mutation. The generation time of *E. coli* per day in the human gut has been estimated to be between 6 and 20 generations per day (*Ghalayini et al., 2018*). For large plasmids that exist at a copy number of ~1, the plasmid generation time is the cell generation time. More generally, for a plasmid copy number $p$ the number of replications of the plasmid expected for a given number of cell generations $g$ will be $p \times g$ (assuming that plasmid copies are simply and linearly related to the realised number of replications per cell). A crude estimate for the expected mutation rate per time period for a plasmid is therefore given by $L \times m \times p \times g$. For a plasmid of $L{=}100$ kbp and $P{=}1$, assuming $m{=}[0.1{-}1] \times 10^{-9}$ per site per generation and $g{=}[6{-}20]{\times}365$ per year, one would expect it to accumulate ~0.5 mutations a year (between ~0.02 and 0.7 depending on assumptions). One obtains the same result for $L{=}10$ kbp and $P{=}10$. There is a strong inverse correlation between plasmid size and copy number. This suggests that a suitable upper bound for the expected number of mutations for a typical plasmid per year (under neutral evolution) is of the order of magnitude of 1 SNP a year. This rough 'SNPs and years' rule-of-thumb appears consistent with known empirical results. For example: 100 kbp I1-type *Shigella* plasmids isolated between 2007 and 2010 in Vietnam were separated by at most 2 SNPs (*Holt et al., 2013*); 30 kbp X4-type plasmids carrying *mcr-1* isolated between 2016 and 2018 in China were separated by most 4 SNPs (*Shen et al., 2020*) (analysis not shown); 63.5 kbp pOXA48-like plasmids ($n{=}202$) in *K. pneumoniae* collected across Europe between 2013 and 2014 as part of EUSCAPE were overwhelmingly within 2 SNPs of each other (176/202) (*David et al., 2020*); the same was true of 45.4 kbp IncX3 plasmids ($n{=}135$) from the EUSCAPE dataset (all were within 6 SNPs of each other; see Figure 4 of that paper); and also of 113.4 kbp pKpQIL-like plasmids

(*n=91*) from the EUSCAPE dataset – although a minority of these plasmids were separated by up to 20 SNPs, which seems suggestive of either ancestry before the 2-year sampling frame or recombination.

### Pangraph analysis

We used pangraph (*Noll et al., 2022*) (v. 0.5.0) to build a pangraph of the clade within plasmid cluster 2, using the `--circular` flag and otherwise default parameters. We removed duplicated blocks from the pangraph. We used pangraph export (`--edge-minimum-length 0`, default parameters) to export the graph to GFA format and then visualised this using Bandage (*Wick et al., 2015*). *Supplementary file 4* used Prokka annotations (see above) of the core and accessory pancontigs.

### Data visualisation

Plots were primally produced using the R library ggplot2 (*Gómez-Rubio, 2017*), with additional graphics in BioRender (*Munday, 2021*).

## Acknowledgements

This work was funded by the Antimicrobial Resistance Cross-council Initiative supported by the seven research councils (grant NE/N019989/1). The UKCEH component of the REHAB consortium was supported by the Natural Environment Research Council (NERC) (grant NE/N019660/1). DWC, SG, TEAP, and NS are supported by the National Institute for Health Research Health Protection Research Unit (NIHR HPRU) in Healthcare-Associated Infections and Antimicrobial Resistance at the University of Oxford in partnership with Public Health England (PHE) (grant HPRU-2012–10,041 and NIHR200915). DWC and TEAP are also supported by the NIHR Oxford Biomedical Research Centre. The computational aspects of this research were funded from the NIHR Oxford BRC with additional support from a Wellcome Trust Core Award Grant (grant 203141/Z/16/Z). The views expressed are those of the authors and not necessarily those of the NHS, the NIHR, the Department of Health or Public Health England. WM and KKC are supported by a scholarship from the Medical Research Foundation National PhD Training Programme in Antimicrobial Resistance Research (MRF-145-0004-TPG-AVISO). NS is an Oxford Martin Fellow and a Senior NIHR BRC Oxford Fellow. LPS is a Sir Henry Wellcome Postdoctoral Fellow funded by Wellcome (grant 220422/Z/20/Z). This study was funded by the Antimicrobial Resistance Cross-council Initiative supported by the seven research councils and the NIHR, UK.

## Additional information

#### Group author details

**REHAB Consortium**

**Manal AbuOun**: Animal and Plant Health Agency,Weybridge, Addlestone, United Kingdom; **Muna F Anjum**: Animal and Plant Health Agency,Weybridge, Addlestone, United Kingdom; **Mark J Bailey**: UK Centre for Ecology & Hydrology, Wallingford, United Kingdom; **Brett H**: Thames Water Utilities, Reading, United Kingdom; **Mike J Bowes**: UK Centre for Ecology & Hydrology, Wallingford, United Kingdom; **Kevin K Chau**: Nuffield Department of Medicine, University of Oxford, Oxford, United Kingdom; **Derrick W Crook**: Nuffield Department of Medicine, University of Oxford, Oxford, United Kingdom; NIHR HPRU in healthcare-associated infection and antimicrobial resistance, University of Oxford, Oxford, United Kingdom; NIHR Oxford Biomedical Research Centre, University of Oxford, Oxford, United Kingdom; **Nicola de Maio**: Nuffield Department of Medicine, University of Oxford, Oxford, United Kingdom; **Nicholas Duggett**: Animal and Plant Health Agency,Weybridge, Addlestone, United Kingdom; **Daniel J Wilson**: Nuffield Department of Medicine, University of Oxford, Oxford, United Kingdom; Wellcome Trust Centre for Human Genetics, University of Oxford, Oxford, United Kingdom; **Daniel Gilson**: Animal and Plant Health Agency,Weybridge, Addlestone, United Kingdom; **H Soon Gweon**: UK Centre for Ecology & Hydrology, Wallingford, United Kingdom; University of Reading, Reading, United Kingdom; **Alasdair Hubbard**: Department of Tropical Disease Biology, Liverpool School of Tropical Medicine, Liverpool, United Kingdom; **Sarah J Hoosdally**: Nuffield Department of Medicine, University of Oxford, Oxford, United Kingdom; **William Matlock**:

Nuffield Department of Medicine, University of Oxford, Oxford, United Kingdom; **James Kavanagh**: Nuffield Department of Medicine, University of Oxford, Oxford, United Kingdom; **Hannah Jones**: Animal and Plant Health Agency,Weybridge, Addlestone, United Kingdom; **Timothy EA Peto**: Nuffield Department of Medicine, University of Oxford, Oxford, United Kingdom; NIHR HPRU in healthcare-associated infection and antimicrobial resistance, University of Oxford, Oxford, United Kingdom; NIHR Oxford Biomedical Research Centre, University of Oxford, Oxford, United Kingdom; **Daniel S Read**: UK Centre for Ecology & Hydrology, Wallingford, United Kingdom; **Robert Sebra**: Icahn Institute of Data Science and Genomic Technology, Mt Sinai, New York, United States; **Liam P Shaw**: Nuffield Department of Medicine, University of Oxford, Oxford, United Kingdom; **Anna E Sheppard**: Nuffield Department of Medicine, University of Oxford, Oxford, United Kingdom; NIHR HPRU in healthcare-associated infection and antimicrobial resistance, University of Oxford, Oxford, United Kingdom; **Richard P Smith**: Animal and Plant Health Agency,Weybridge, Addlestone, United Kingdom; **Emma Stubberfield**: Animal and Plant Health Agency,Weybridge, Addlestone, United Kingdom; **Nicole Stoesser**: Nuffield Department of Medicine, University of Oxford, Oxford, United Kingdom; NIHR HPRU in healthcare-associated infection and antimicrobial resistance, University of Oxford, Oxford, United Kingdom; NIHR Oxford Biomedical Research Centre, University of Oxford, Oxford, United Kingdom; **Jeremy Swann**: Nuffield Department of Medicine, University of Oxford, Oxford, United Kingdom; **A Sarah Walker**: Nuffield Department of Medicine, University of Oxford, Oxford, United Kingdom; NIHR HPRU in healthcare-associated infection and antimicrobial resistance, University of Oxford, Oxford, United Kingdom; NIHR Oxford Biomedical Research Centre, University of Oxford, Oxford, United Kingdom; **Neil Woodford**: Antimicrobial Resistance and Healthcare Associated Infections (AMRHAI) Reference Unit, National Infection Service, Public Health England, London, United Kingdom

## Funding

| Funder | Grant reference number | Author |
| --- | --- | --- |
| Medical Research Foundation | MRF-145-0004-TPG-AVISO | William Matlock Kevin K Chau |
| Antimicrobial Resistance Cross-council Initiative | NE/N019989/1 | REHAB Consortium |
| Natural Environment Research Council | NE/N019660/1 | REHAB Consortium |
| Public Health England | HPRU-2012–10041 | Derrick W Crook |
| Public Health England | NIHR200915 | Derrick W Crook |
| Wellcome Trust | 203141/Z/16/Z | REHAB Consortium |
| Wellcome | 220422/Z/20/Z | Liam P Shaw |

The funders had no role in study design, data collection and interpretation, or the decision to submit the work for publication. For the purpose of Open Access, the authors have applied a CC BY public copyright license to any Author Accepted Manuscript version arising from this submission.

## Author contributions

William Matlock, Conceptualization, Data curation, Software, Formal analysis, Validation, Investigation, Visualization, Methodology, Writing – original draft, Writing – review and editing; Samuel Lipworth, Data curation, Formal analysis; Kevin K Chau, James Kavanagh, Data curation; Manal AbuOun, Leanne Barker, Data curation, Methodology; Monique Andersson, Sarah Oakley, Marcus Morgan, Project administration; Derrick W Crook, Resources, Supervision; Daniel S Read, Conceptualization, Resources, Data curation, Supervision; Muna Anjum, Conceptualization; Liam P Shaw, Conceptualization, Formal analysis, Supervision, Writing – original draft, Writing – review and editing; Nicole Stoesser, Conceptualization, Supervision, Writing – original draft, Writing – review and editing; REHAB Consortium, Conceptualization, Funding acquisition, Project administration, Resources, Supervision

## Author ORCIDs

William Matlock (iD) http://orcid.org/0000-0001-5608-0423

Derrick W Crook  http://orcid.org/0000-0002-0590-2850

Nicole Stoesser  http://orcid.org/0000-0002-4508-7969

**Decision letter and Author response**

Decision letter https://doi.org/10.7554/eLife.85302.sa1

Author response https://doi.org/10.7554/eLife.85302.sa2

## Additional files

### Supplementary files

• MDAR checklist

• Supplementary file 1. In order, this file contains (i) a metadata table for all plasmid clusters where every plasmid has at least one core gene. Presented are the cluster name, cluster size, mlst PubMLST genera of plasmid hosts, plasmid PlasmidFinder annotations, and plasmid NCBIAMRFinder annotations, and (ii) a core gene phylogeny and consensus gene synteny heatmap for all clusters in the metadata table, in cluster size decreasing order. Core gene phylogeny scales are in single nucleotide polymorphisms (SNPs).

• Supplementary file 2. Fritz and Purvis' D estimates for the *n=27/62* plasmid clusters that contained both bloodstream infection (BSI) and livestock-associated plasmids.

• Supplementary file 3. Isolate and assembly metadata.

• Supplementary file 4. Cluster 2 core and accessory pancontig gene annotations.

### Data availability

REHAB reads and assemblies and assemblies can be found in the NCBI BioProject PRJNA605147. BSI reads and assemblies can be found at https://doi.org/10.25452/figshare.plus.24573268. Analysis scripts can be found in the GitHub repository https://github.com/wtmatlock/oxfordshire-overlap (copy archived at *Matlock, 2023*).

The following dataset was generated:

| Author(s) | Year | Dataset title | Dataset URL | Database and Identifier |
|---|---|---|---|---|
| Matlock WM, Lipworth S, Shaw LP, Stoesser N | 2023 | Oxfordshire human bloodstream infection isolate sequence data | https://doi.org/10.25452/figshare.plus.24573268 | Figshare+, 10.25452/figshare.plus.24573268 |

The following previously published datasets were used:

| Author(s) | Year | Dataset title | Dataset URL | Database and Identifier |
|---|---|---|---|---|
| Shaw LP, Chau KK, Kavanagh J, AbuOun M, Stubberfield E, Gweon HS | 2021 | REHAB | https://www.ncbi.nlm.nih.gov/bioproject/PRJNA605147 | NCBI BioProject, PRJNA605147 |
| Lipworth S, Vihta KD, Chau K, Barker L, George S, Kavanagh J | 2021 | BSI | https://www.ncbi.nlm.nih.gov/bioproject/PRJNA604975 | NCBI BioProject, PRJNA604975 |

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

## Appendix 1

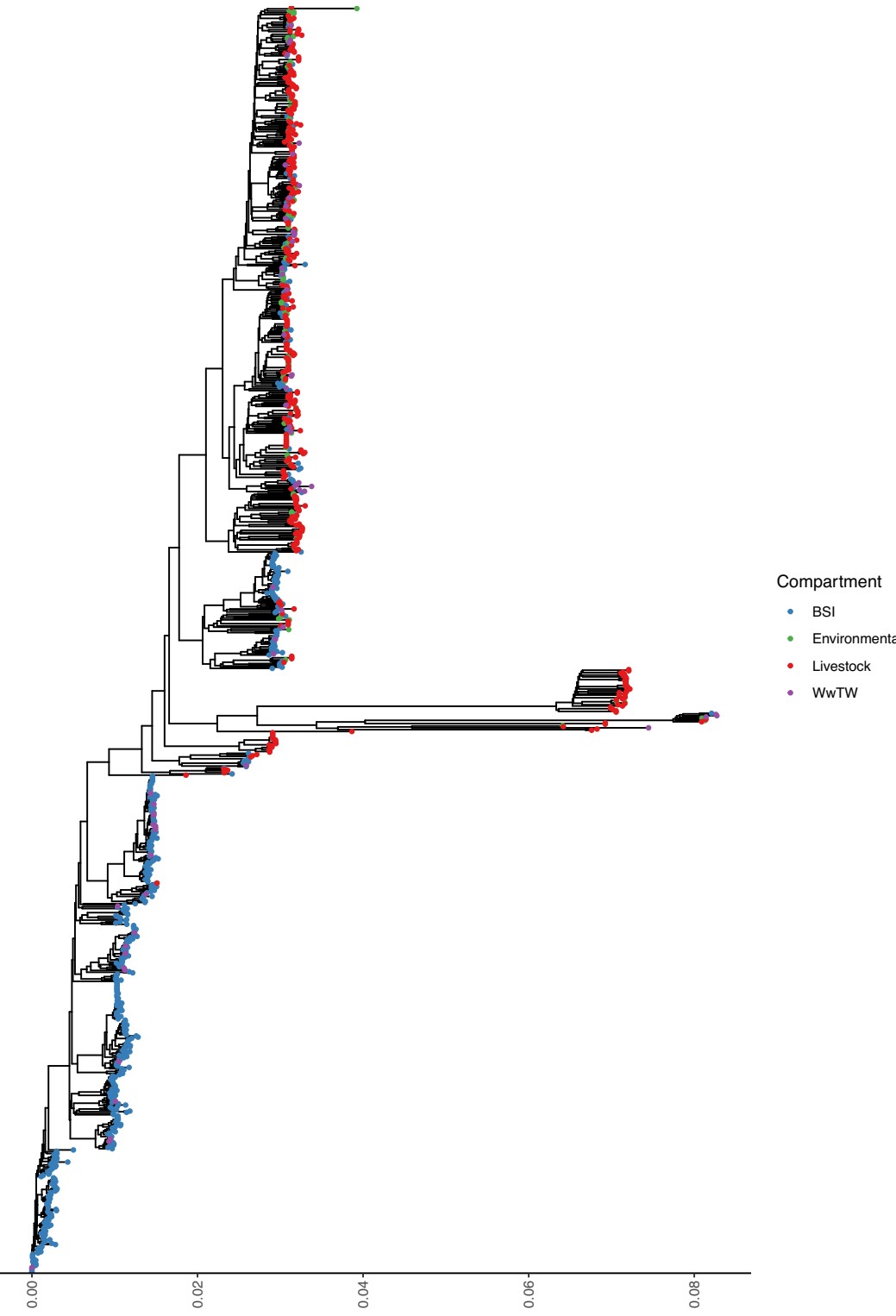

**Appendix 1—figure 1.** Mash tree for *n=1044 E. coli* chromosomes. Tree tips are coloured by sampling compartment, scale is Mash distance.

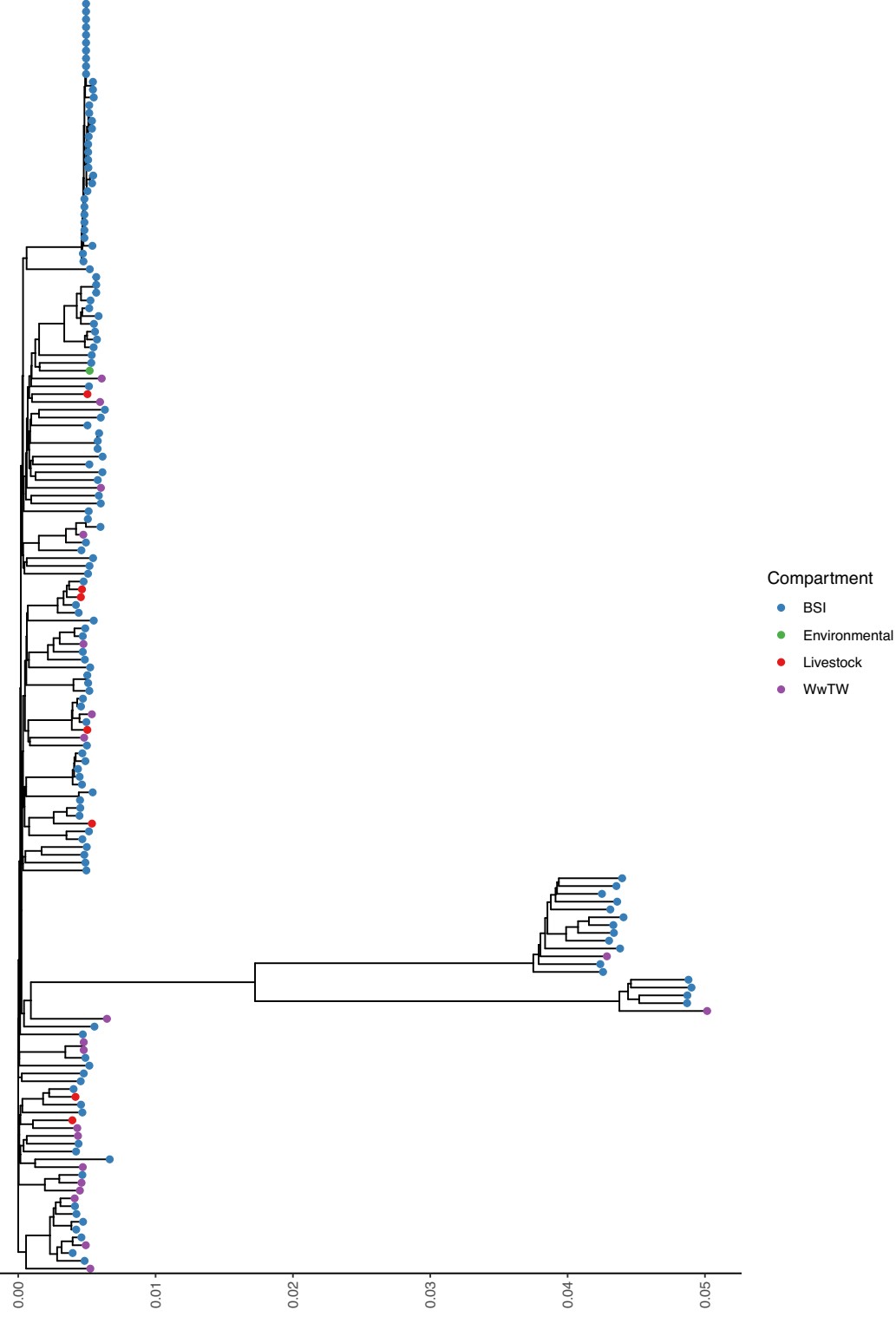

**Appendix 1—figure 2.** Mash tree for *n=163 K. pneumoniae* chromosomes. Tree tips are coloured by sampling compartment, scale is Mash distance.

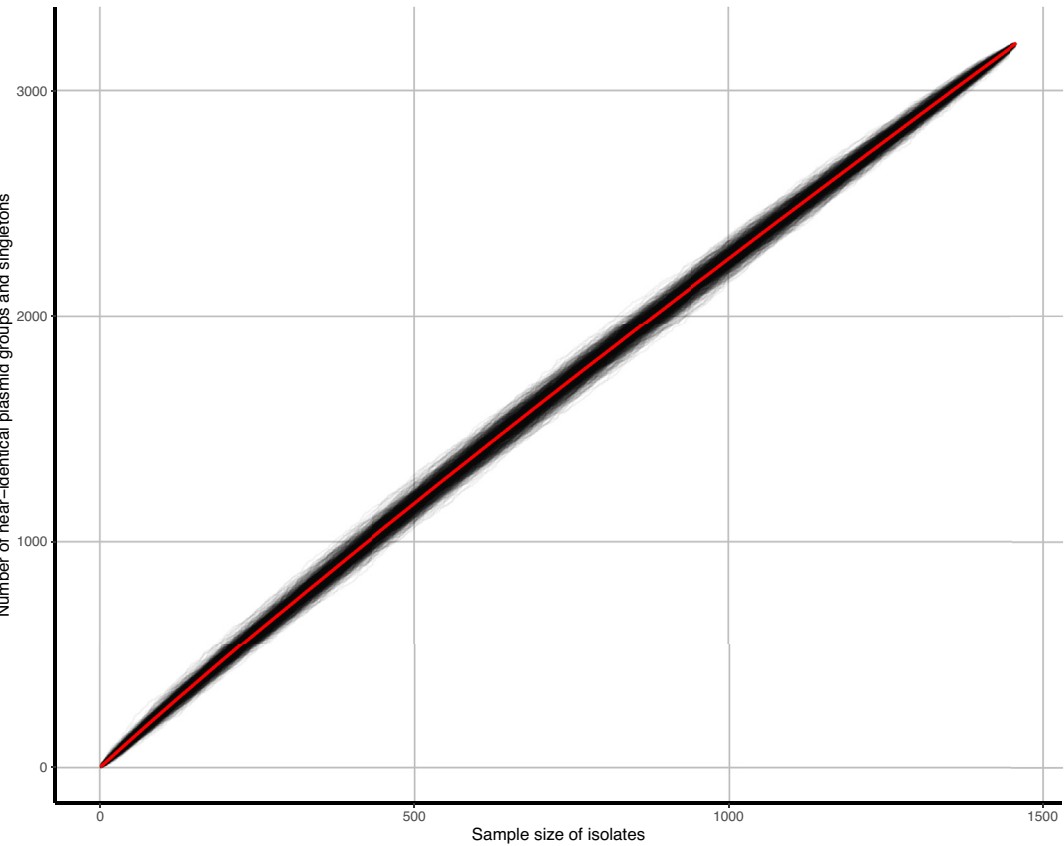

**Appendix 1—figure 3.** Accumulation curves of near-identical plasmid groups and singletons against isolate sample size. Black lines represent *b*=1000 bootstrap simulations, the red line represents their average.

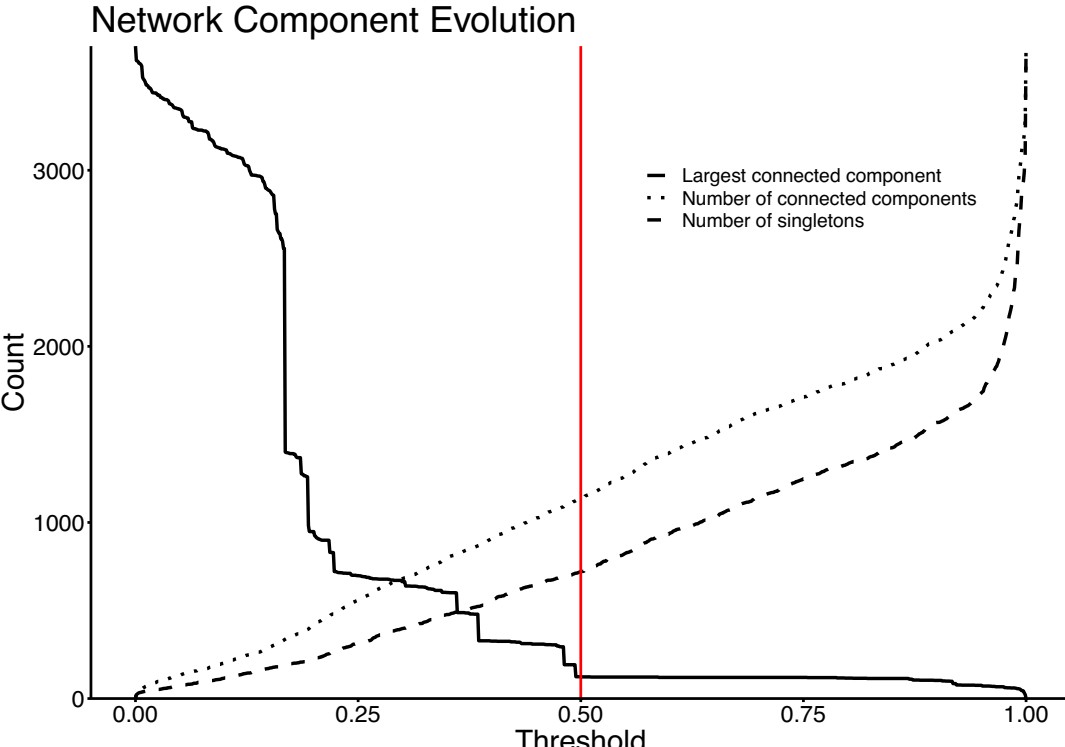

**Appendix 1—figure 4.** Network evolution of largest connected component, number of connected components, and number of singletons, as edges are removed at increasing JI thresholds. The vertical red line represents the chosen threshold of JI=0.5.

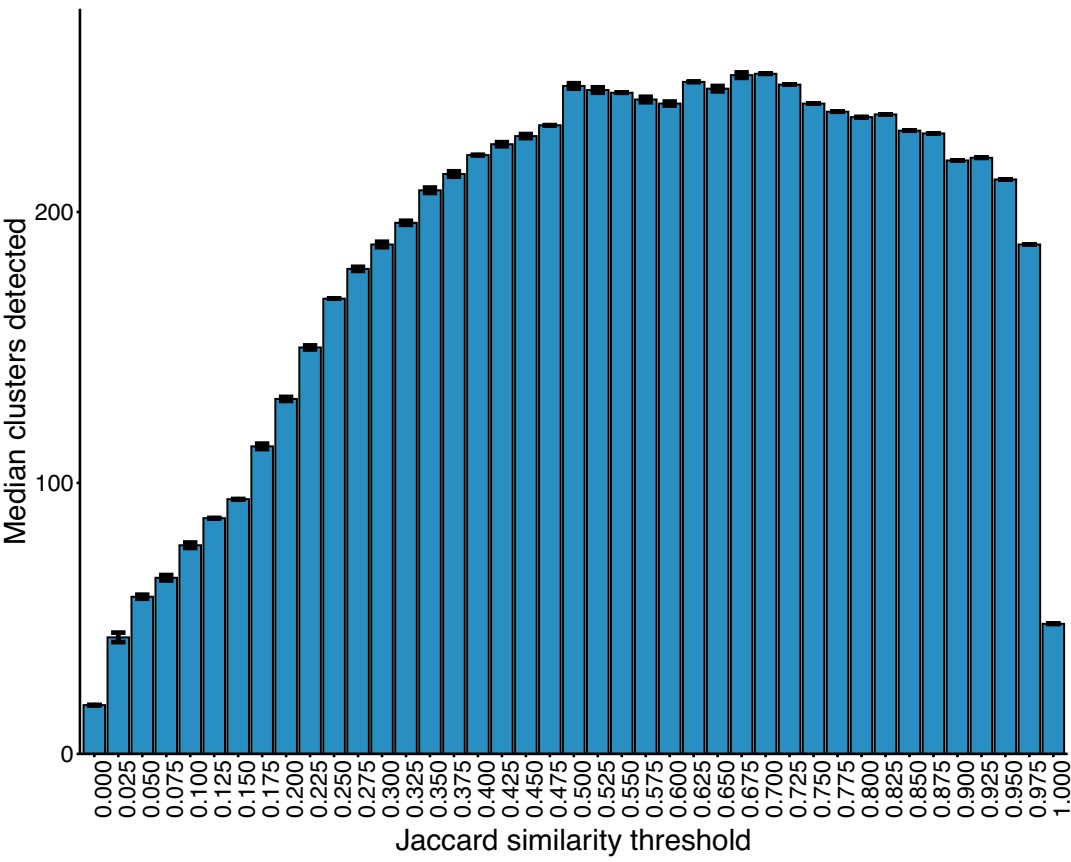

**Appendix 1—figure 5.** Number of clusters detected within the plasmid network at increasing JI thresholds. Interval bars represent the IQR in cluster number at a given threshold over 50 runs of the Louvain algorithm.

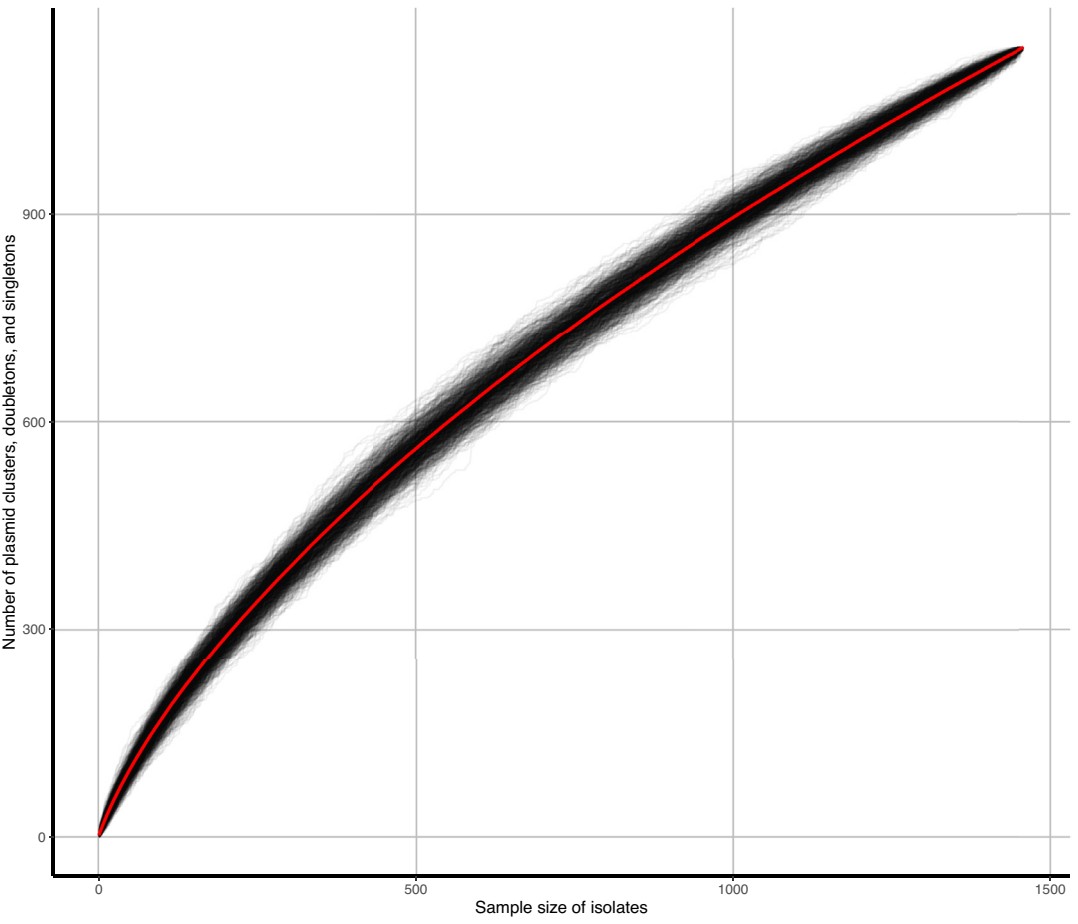

**Appendix 1—figure 6.** Accumulation curves of plasmid clusters, doubletons, and singletons against isolate sample size. Black lines represent b=1000 bootstrap simulations, red line represents their average.

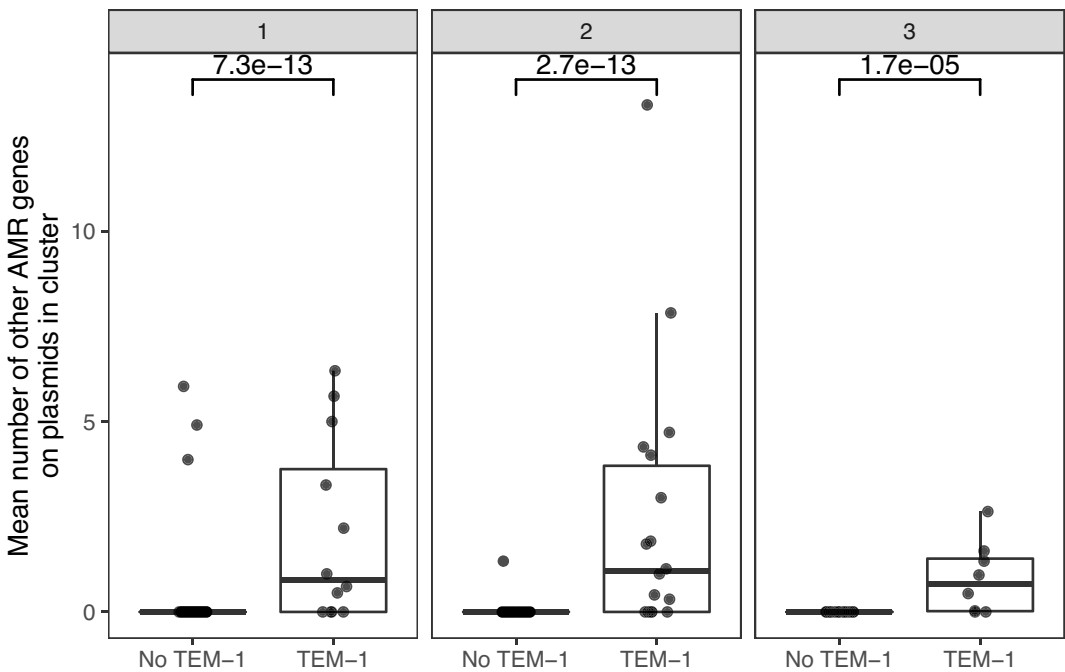

**Appendix 1—figure 7.** Plasmid clusters containing blaTEM-1 carry more AMR genes. Each point is one plasmid cluster. *n=247* clusters are shown, with panels facetted by the number of niches the plasmid cluster represented. p-values are from the Wilcoxon test.

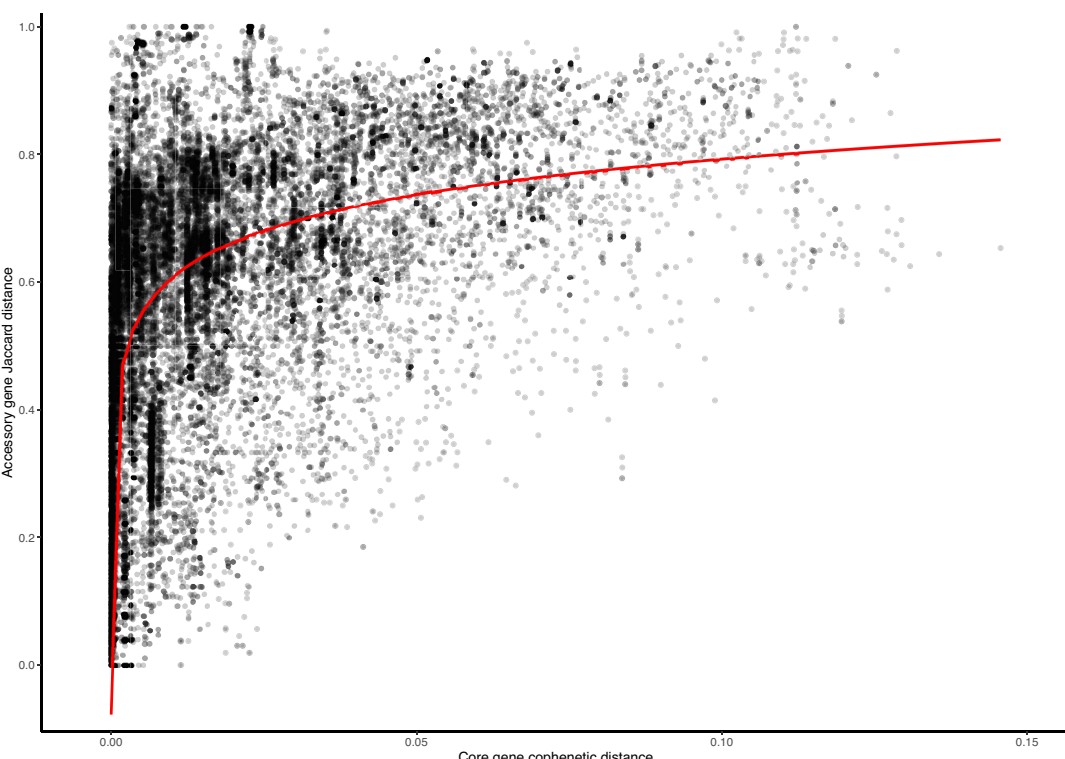

**Appendix 1—figure 8.** Plasmid accessory gene presence/absence Jaccard distance against core-gene cophenetic distance. Presented are data points from 27/247 clusters for which (**i**) all plasmids had at least 1 core gene, and (ii) the cluster contained at least 50 accessory genes. The red line is a statistically significant (p-value <2.2e-16) log-transformed linear regression.

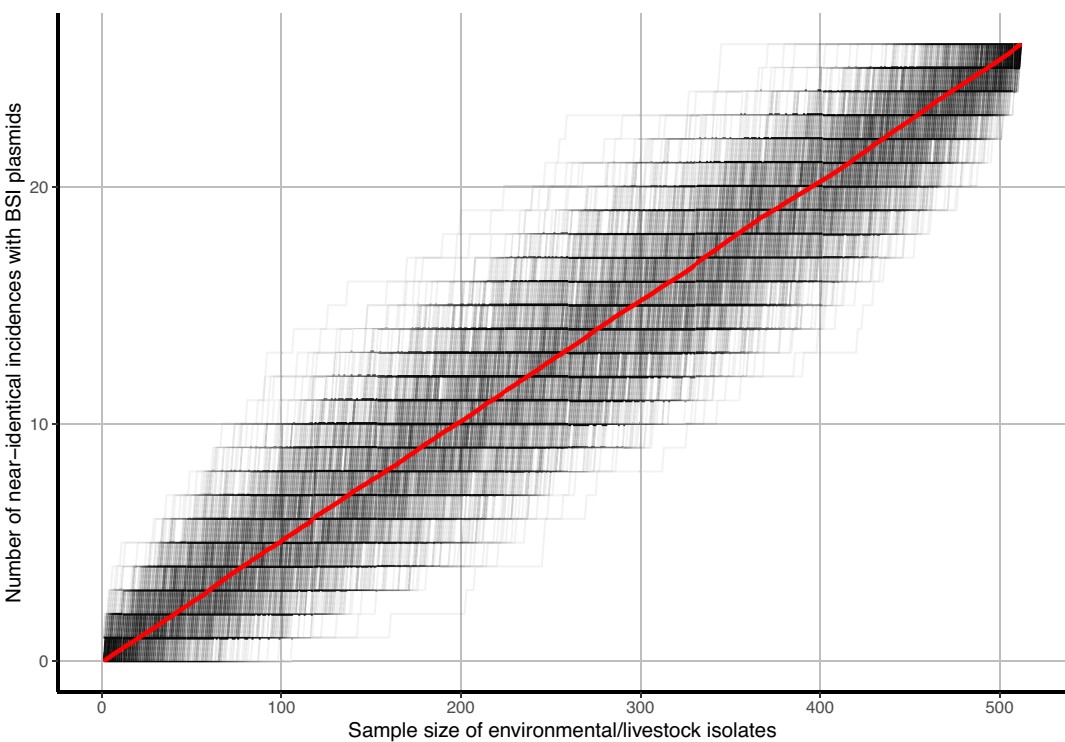

**Appendix 1—figure 9.** Accumulation curves of near-identical plasmid matches with bloodstream infection (BSI) plasmids and singletons against livestock-associated (environmental soils/livestock) isolate sample size. Black lines represent *b*=1000 bootstrap simulations, red line represents their average.

