## [Editor Report]

This study presents valuable findings on the dissemination of plasmids. The analysis, involving a geographically and temporally restricted collection of fully assembled genomes of 1458 isolates carrying in total of 3697 plasmids representing five major Enterobacterales genera, convincingly demonstrated that similar plasmids were shared between genera, species, and clones, within and between ecological niches. Given the size of the dataset and the very detailed level of analysis, this important study contributes to insights into the flow of plasmids, including those carrying antimicrobial resistance genes, across niches.

---

## [Decision Letter]

**Decision letter after peer review:**

Thank you for submitting your article "Enterobacterales plasmid sharing amongst human bloodstream infections, livestock, wastewater, and waterway niches in Oxfordshire, UK" for consideration by *eLife*. Your article has been reviewed by 2 peer reviewers, including Marc J Bonten as the Reviewing Editor and Reviewer #1, and the evaluation has been overseen by Bavesh Kana as the Senior Editor. The following individual involved in review of your submission have agreed to reveal their identity: Rob Willems (Reviewer #2).

Essential revisions:

Page 9; Lines 159-164: It is not clear what the plasmid clustering that is mentioned here represent, because this is only explained on page 10. Therefore, I suggest moving this section to after the explanation of the plasmid clusters.

Page 11; lines 190-196. I suggest to also indicate here the homogeneity (h) and completeness (c) values with respect to sampled niche, so that these plasmid cluster scores can also be evaluated with respect to their ecological characteristics.

Page 12; lines 225-230, The ratio between the size of the core plasmid backbone and plasmid accessory genome as given in the text does not seem to correlate with what is shown in Figure 3e. The authors write in the text that at individual plasmid level core genes shared by a cluster comprised a median proportion of 62,5%, while looking at the ratio core/accessory plasmid genes in the different clusters in Figure 3e this seems to be much smaller, like more in the line of 15-20%. Can the authors explain this?

Page 13, line 251. I suggest replacing "evolution" by "dissemination" in the title of this section, since most of this section describes the inter-host transfer. Plasmid evolution is only being discussed at the end of this section, on page 15, starting at line 290.

Page 15, lines 290-299. It would be interesting, as an example of plasmid evolution in this clade of 44 plasmids (indicated in Figure 4b), to provide more information on which genes, were located on the core blocks (Figure 4e) and/or within the accessory blocks that were gained or lost (Figure 4f) when comparing BSI and livestock plasmids. This would give some more insight into genes involved in the adaptive evolution of plasmids after a host jump.

*Reviewer #1 (Recommendations for the authors):*

I have no specific comments for the authors

*Reviewer #2 (Recommendations for the authors):*

There are a few other comments and suggestions that may improve the manuscript.

Page 9; Lines 159-164: It is not clear what the plasmid clustering that is mentioned here represent, because this is only explained on page 10. Therefore, I suggest moving this section to after the explanation of the plasmid clusters.

Page 11; lines 190-196. I suggest to also indicate here the homogeneity (h) and completeness (c) values with respect to sampled niche, so that these plasmid cluster scores can also be evaluated with respect to their ecological characteristics.

Page 12; lines 225-230, The ratio between the size of the core plasmid backbone and plasmid accessory genome as given in the text does not seem to correlate with what is shown in Figure 3e. The authors write in the text that at individual plasmid level core genes shared by a cluster comprised a median proportion of 62,5%, while looking at the ratio core/accessory plasmid genes in the different clusters in Figure 3e this seems to be much smaller, like more in the line of 15-20%. Can the authors explain this?

Page 13, line 251. I suggest replacing "evolution" by "dissemination" in the title of this section, since most of this section describes the inter-host transfer. Plasmid evolution is only being discussed at the end of this section, on page 15, starting at line 290.

Page 15, lines 290-299. It would be interesting, as an example of plasmid evolution in this clade of 44 plasmids (indicated in Figure 4b), to provide more information on which genes, were located on the core blocks (Figure 4e) and/or within the accessory blocks that were gained or lost (Figure 4f) when comparing BSI and livestock plasmids. This would give some more insight into genes involved in the adaptive evolution of plasmids after a host jump.

---

## [Author Response]

Reviewer #2 (Recommendations for the authors):There are a few other comments and suggestions that may improve the manuscript.Page 9; Lines 159-164: It is not clear what the plasmid clustering that is mentioned here represent, because this is only explained on page 10. Therefore, I suggest moving this section to after the explanation of the plasmid clusters.

Thank you for noticing this error. The clustering mentioned here can be now found in lines 244-252. We have also re-numbered the supplementary figures throughout to account for this re-ordering.

Page 11; lines 190-196. I suggest to also indicate here the homogeneity (h) and completeness (c) values with respect to sampled niche, so that these plasmid cluster scores can also be evaluated with respect to their ecological characteristics.

We agree that this would improve the interpretation of the clustering. We have included the homogeneity and completeness scores for sampling niche on lines 222-224:

“When scoring plasmid clusters against broad sampling niche (BSI, livestock-associated or WwTW-associated; Figure 3a), homogeneity was low (*h*=0.12, *c*=0.61), indicating mixed clusters.”

Page 12; lines 225-230, The ratio between the size of the core plasmid backbone and plasmid accessory genome as given in the text does not seem to correlate with what is shown in Figure 3e. The authors write in the text that at individual plasmid level core genes shared by a cluster comprised a median proportion of 62,5%, while looking at the ratio core/accessory plasmid genes in the different clusters in Figure 3e this seems to be much smaller, like more in the line of 15-20%. Can the authors explain this?

We appreciate that this was unclear. The ‘median 62.5% proportion’ statistic (line 246) refers to the median proportion of genes within each plasmid which were core genes, as determined by the cluster pangenome analysis, and shown by the boxplots in Figure 3e. The bar chart in Figure 3e shows the proportion of unique genes in the entire cluster pangenome which were core or accessory. This distinction reflects that whilst the entire cluster pangenome contains far more accessory genes than core genes, plasmids have many core genes in common. We have amended the caption of Figure 3e to better clarify the distinction (lines 1027-1029):

“(e) Plasmid core and accessory genomes: Left hand axis shows the count of core and accessory coding sequences (CDS) for the entire cluster as a bar chart. Right hand axis shows plasmid core gene proportions (i.e., core CDS/total CDS for each plasmid) as a boxplot.”

Page 13, line 251. I suggest replacing "evolution" by "dissemination" in the title of this section, since most of this section describes the inter-host transfer. Plasmid evolution is only being discussed at the end of this section, on page 15, starting at line 290.

We have replaced “evolution” by “dissemination” (line 291).

Page 15, lines 290-299. It would be interesting, as an example of plasmid evolution in this clade of 44 plasmids (indicated in Figure 4b), to provide more information on which genes, were located on the core blocks (Figure 4e) and/or within the accessory blocks that were gained or lost (Figure 4f) when comparing BSI and livestock plasmids. This would give some more insight into genes involved in the adaptive evolution of plasmids after a host jump.

We agree that more information on core/accessory annotations would provide greater insight into the genes involved in potential adaptation, and a more detailed evaluation of this would represent interesting future work. We have added these details (lines 339-346), including a complete list of annotations in Supp. File 5:

“Core and accessory pancontigs contained (22%; 57/261) and (78%; 204/261) of gene annotations, respectively, of which over half encoded hypothetical proteins (51%; 134/261; see Supp. File 5 and Materials and methods). Core annotations included replication (*repB*) and conjugation (*finO*, *traI*, *traM*) proteins, whereas accessory gene annotations included antimicrobial resistance (*bcr*, *blaTEM, tetA, tetR*) and mercury resistance (*merA*, *merC*, *merP*, *merT*) proteins. Transposase/insertion sequence annotations were disproportionately found in accessory pancontigs (88%; 38/43) versus core pancontigs (12%; 5/43).”